# Photogrammetric approach to detect road pavement friction

**Zdeněk Svatý**[1‡], **Pavel Vrtal**[1*], **Roman Shults**[2], **Tomáš Kohout**[1], **Luboš Nouzovský**[1], **Tomáš Blodek**[1], **Karel Kocián**[1]

1 CTU in Prague, FTS, Department of Forensic Experts in Transportation, Prague 1, The Czech Republic, 2 King Fahd University of Petroleum & Minerals, Interdisciplinary Research Center for Aviation and Space Exploration, Dhahran, Saudi Arabia

‡ Zdeněk Svatý–Main author. Other authors contributed equally to this work.
* vrtalpav@fd.cvut.cz

## Abstract

This paper deals with evaluation of photogrammetric measurements of pavement macro-texture for the purposes of new non-contact method for determination of pavement skid resistance. The aim was to create a method that yields results comparable to existing pavement friction measurement techniques, particularly the sand method, and to assess whether the measurements obtained using the non-contact approach are equally effective. Additionally, the goal was to introduce and define limits for camera calibration applicable to similar cases of object measurements at close distances. Based on existing standards, specific pavement samples exhibiting various macrotexture qualities were selected. The authors conducted 23 different combinations of photogrammetric measurements based on their findings during the evaluations. A comparison of the applied procedures with a reference measurement using the sand method was made, and the measurement error rate was calculated. By evaluating the different variations of the road surface models created, it was concluded that the most reliable results for defining friction are obtained by inter-leaving the test specimen through a plane based on the principle of best fit using the four highest points of the macrotexture. The ideal size of the sub-areas for applying the mask is 7.5–10 mm. Photogrammetry has proven to be a suitable and accurate tool for assessing pavement surface texture and can be considered as a suitable alternative to other skid resistance measurement approaches.

## I. Introduction

The main purpose of a transportation system is to facilitate safe and efficient movement of people and goods. Even though the accident rate has been dropping recently and safety measures have been improving, it can be said that the figures as for consequences of these crashes are still too high. One of the main factors affecting the road safety is the condition of the road, especially, the characteristics of road surface. Pavement surface texture determines the quality of a ride (smoothness, generated noise) together with affecting the overall durability of pavement (deformation, aggregate segregation). The texture measurements are crucial for proper pavement design and management. Moreover, there is evidence [1,2] showing an increased number of traffic accidents due to low skid resistance caused by insufficient pavement texture.

**Data availability statement:** Data related to the final analyses can be found in OSF at https://osf.io/98vkz/ (DOI 10.17605/OSF.IO/98VKZ).

**Funding:** The author(s) received no specific funding for this work.

**Competing interests:** The authors have declared that no competing interests exist.

The main goal of this study is to address the close-range photogrammetry application to generate the point cloud of the test surface. This process is unique, but it is possible to achieve the desired results. Photogrammetry provides accurate and detailed information about surfaces with a well-defined structure. Even relatively smooth surfaces have quite well structure. This fact allows applying image processing techniques to retrieve points from images automatically [3] where the core of this study is use of photogrammetric approach for capturing the surface of interest from multiple surveying stations to reconstruct a three-dimensional surface. Afterwards is used for mean texture depth (MTD). The mean texture depth can then be calculated to determine the safety of the road surface [4].

## II.  Current knowledge

Currently, the most common and simplest method for assessing road texture is the "sand method" [4]. The measured method is used to determine the mean texture depth (MTD), which determines the total surface depression volume. The test is carried out using $25\,cm^3$ of $0.18$–$0.25\,mm$ glass balls which are poured on the surface to be tested (properly cleaned beforehand). The glass balls are then spread into a circular shape until the entire volume of material is embedded in the depressions of the road and the diameter of the circle no longer increases. The diameter of the circle is then measured in at least four places and the mean macrotexture depth is calculated from the average of the measured values, which is then assigned to one of the five classification levels according to EN 13036-1 [5] for the measurement method.

The level of quality of the skid resistance properties obtained by this method is partly illustrative, since the skid resistance properties also depend on the character of the microtexture of the surface aggregate, which this method is essentially unable to take into consideration. Thus, it may be that a surface with a classification grade of 3 may theoretically have better skid resistance properties (due to a suitable microtexture) than a surface with a classification grade of 1 whose microtexture is unsuitable.

Road texture has been categorized by PIARC [6] into three ranges by its wavelength and amplitude, or space frequency respectively: microtexture with a relative wavelength of $<0.5\,mm$, macro-texture with a relative wavelength between $1\,mm$ and $50\,mm$ and megatexture with a relative wavelength between $50\,mm$ and $500\,mm$. Generally, the range of texture depth is a basis for a current international classification scale of pavement skid resistance. This classification has 5 levels, from excellent to hazardous. Also, the macro-texture helps to prevent skidding as vehicle tires transverse the pavement structure. However, this ability is adversely affected by surface water, which can lead to significant reductions of macrotexture [7,8].

As an example of comparing existing methods with new methods, the American study can be used. It compares the results of two measurement methods for pavement surface macro-texture: the ASTM E965 sand patch volumetric method and the Digital Surface Roughness Meter (DSRM, laser technique) conducted at a total of 13 different paved road sites in metropolitan Clark County, Nevada. Least-squares regression of DSRM-measured mean profile depth (MPD) on sand patch mean texture depth (MTD) was statistically significant ($r^2 = 0.95$, $p < 0.05$), with DSRM estimates within $0.1\,mm$ of sand patch in 58 of 61 cases. The DSRM method is able to estimate MTD with a higher level of accuracy ($0.08\,mm$) than the sand method ($0.21\,mm$) and sample multiple locations in an inaccessible site in the same time. [9]

Another method used to measure surface properties is the so-called Circular Texture Meter (CTMeter), which measures the mean road surface profile depth (MPD). The circle profile is divided into eight equal sections. Two of these are approximately parallel to the direction of travel and two are approximately perpendicular to the direction of travel. The remaining four are approximately ± 45 degrees to the direction of travel. This method uses a transparent perpendicular cylinder that is situated on a rubber pad placed on the roadway. At the bottom

of the cylinder there is a valve which is closed and the cylinder is filled with water. The valve is then opened and the time (OFT) for the water level to drop by a fixed value is measured. Data from three years of testing were used to evaluate the correlation of the mean profile depth obtained by the CTMeter with the volumetric mean texture depth and runoff time. In all cases the correlation coefficients were very high. However, follow-up testing is necessary to verify the results for porous surfaces [10].

The following study deals with a non-contact method for evaluating the skid resistance (SR) of a roadway based on 3D textures. In the measurement, the point cloud was scanned using a stationary laser scanner. Before data collection, four positioning targets were placed around the measured sample to determine the relative coordinates. During data collection, the scanner was moved around the sample to ensure that the road surface points were completely captured. In accordance with this measurement, the frictional performance for each position of interest was estimated using the traditional British Pendulum Test (BPT), where the British Pendulum Number (BPN) is measured. The correlation between the 3D texture data and the BPN was determined using a Convolutional Neural Network (CNN) model, the output of which was the friction characterization parameters. The results of this method showed that textures with wavelengths above 2.40 mm are crucial for wet friction. The newly developed non-contact method shows high feasibility in predicting shear resistance and effectively controls the relative error within 14% [11].

A similar study also deals with the assessment of pavement skid resistance (SR) using an automated image-based system via Wavelet Transform (WT), which consists of distinct modules including pre-processing, feature extraction, approximate indexes in three different directions (horizontal, vertical, and diagonal), and the overall index. This method can be used to obtain texture information. Images were first acquired by an automated image acquisition system (IAS) and then an image processing system and a knowledge-based decision support system (DSS) were designed. The method was validated on a database of road surface images taken in dry and wet conditions. The practical tests conducted in this study have demonstrated the advantages of the system, including its fast processing, ease of operation and user-friendly features [12].

The last mentioned study, similar to the presented article, focuses on the measurement of macrotexture based on multi-view images taken with a smartphone with a typical resolution of 12 megapixels (specifically the iPhone 8 Plus) using a conventional wide-angle camera. Using the captured images, accurate and dense 3D point clouds of the roadway are generated using SFM (structure-from-motion) and MVS (multi-view stereo) techniques. The 3D point clouds were generated from 790 images taken at 31 different locations. To evaluate and compare the measurements of this image-based method, an Ames laser texture scanner (LTS) was used. The smartphone images with a homemade control frame were shown to be sufficient to reconstruct accurate high-resolution 3D point clouds. The results of this study show that the MPD measured by the laser texture scanner and the smartphone images differ by an average of 4% and a maximum of 9%. The quality of this method can be ensured by taking approximately 30 images for a single point and selecting up to eight points on the roadway. [13]

All the above publications and measurement procedures are suitable supporting material for this article.

## III. Method

The method of work was divided into the following parts, which are described in the diagram below (Fig 1):

The photogrammetric approach requires the assignment of the spatial reference coordinate system and a set of reference points with known coordinates. To fulfill these requirements,

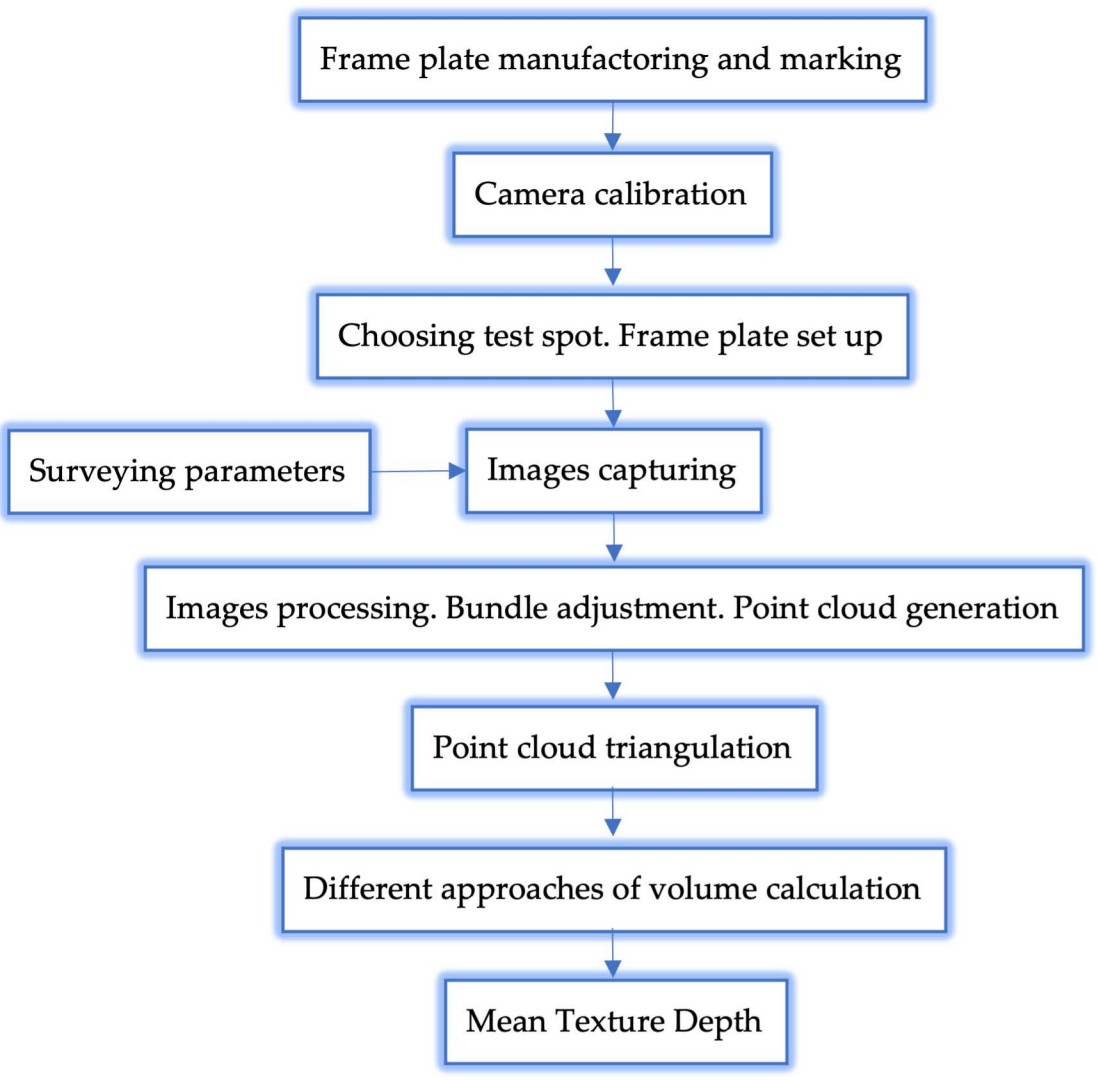

**Fig 1. Photogrammetric approach workflow for MTD calculation.**

a particular frame plate was suggested and manufactured. This frame plate is made of steel, and in the middle of the plate is a rectangular hole with $110 \times 110$ mm dimensions. The hole size was selected regarding the radius of the patch in the SPM and respective standards [5,14]. There are 77 targets in total were marked on the plate. Among the whole targets' set, 56 targets are coded targets. For this reason, the images may be processed automatically in various photogrammetric software with automatic recognition of coded targets. The rest of the targets are simple circled targets (Fig 2).

The coordinates of the targets were determined by the hand-held scanner (REVscan–Handyscan 3D) with a root mean square error of 0.05 mm [15]. The frame's coordinate system is right-handed with xy plane that coincides with the plate surface and a z-axis perpendicular to that surface. The image capturing for each test spot was carried out using a non-metric consumer-grade digital camera NIKON D7500 [16].

Regarding the workflow, after the frame plate manufacturing, the necessary step is camera calibration. The typical camera distortion model that may be applied is the Brown model [17],

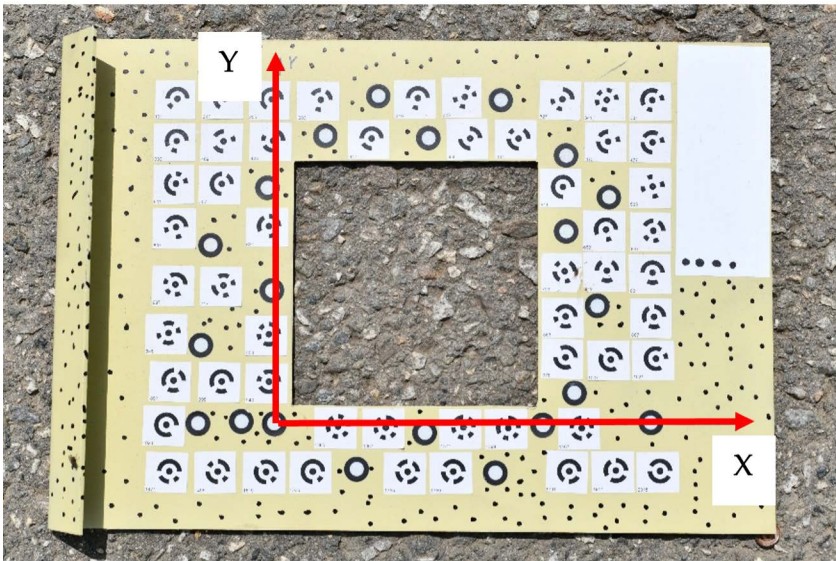

**Fig 2. Frame plate with coded and circled targets.**

or the polynomial model for radial distortion, neglecting tangential distortion. The test spot choosing and its preparation for measurements are performed according to [18]. The calculation of surveying parameters is an indispensable part of photogrammetric surveying. Since the surveying distances are extremely short (up to 1 m), accounting for the field of depth and defocusing errors is mandatory. Images gathering is carried out in a line of general requirements for a photogrammetric survey, namely: a number of images from 12 up to 24; optical axis inclination regarding the frame plate non-less than 45°; test spot coverage on any image non-less than 80%; for a suitable bundle geometry at least 4–5 images must be taken with rotation on 90°; the typical surveying distance 0.5–0.7 m.

Further image processing can be performed using various photogrammetric software developed specifically for close-range applications (Photomodeler, Agisoft Metashape, Pix4D, etc.) [19–21]. The output of the processing is a point cloud of the test spot. The point cloud triangulation is being carried out in Matlab under the running of the developed code. For that code, the different cases of volume calculation were tested. The final step is the MTD calculation [5].

The surveying parameters are considered the values that ensure the quality of the final results. It means that the parameters provide the necessary surveying conditions. The most important parameter is the surveying distance $L$. The correct distance provides the necessary accuracy for road surface reconstruction. This accuracy depends on the volume accuracy determination. The volume accuracy depends on the particular accuracy of the coordinate determination. Therefore, it is necessary to determine the allowable coordinate errors and their dependence on distance. To find this distance, let us analyse the expression for MTD calculation [22]:

$$MTD_{circle} = \frac{4V}{\pi D^2} \tag{1}$$

However, in this case is more straightforward expression:

$$MTD_{square} = \frac{V}{A}, \tag{2}$$

where *V* is the volume, and *A* is the area of the square. To find the allowable error, one may afford some approximations. The area is accepted as errorless. The volume can be considered as a product of three dimensions.

$$MTD_{square} = \frac{xyz}{A} \tag{3}$$

Under this assumption, one may apply the general law of the error theory and variate the expression to obtain the volume error [22]:

$$\delta MTD_{square} = \frac{\sqrt{\left(yz\delta x\right)^2 + \left(xz\delta y\right)^2 + \left(xy\delta z\right)^2}}{A} \tag{4}$$

where $\delta x, \delta y, \delta z$ are the errors of a point coordinate determination along the axis. The expression can be simplified if we accept that the errors $\delta x, \delta y, \delta z$ are equal to each other,

$$\delta x = \delta y = \delta z = \delta \tag{5}$$

and consequently:

$$\delta MTD_{square} = \frac{\delta\sqrt{\left(yz\right)^2 + \left(xz\right)^2 + \left(xy\right)^2}}{A}, \Rightarrow \delta = \frac{A\delta MTD_{square}}{\sqrt{\left(yz\right)^2 + \left(xz\right)^2 + \left(xy\right)^2}} \tag{6}$$

This expression (6) can be used to estimate the accuracy of the point coordinate determination. Most of the parameters are fixed: $A = 10000 \ mm^2$, $x = 100 \ mm$, $y = 100 \ mm$. Accuracy will therefore vary mainly depending on the depth of the road texture. So, finally, is possible to infer that the main contribution has the error along the *z* axis, and therefore, we may approximate the accuracy of MTD determination $\delta MTD_{square}$ is approximately equal to $\delta z$. Therefore, in the following, the measurement distance can be calculated using the values of $\delta MTD_{square} = \delta z$ values based on standards [22].

Since the correlation between the accuracy of the MTD determination and the accuracy of the point coordinate determination is known, it can be related to the accuracy of the photogrammetric measurement $\delta z$, which is given as [3]:

$$\delta z = \frac{L^2}{Bf}\delta p \frac{1}{\sqrt{n-1}} = \delta MTD_{square} \tag{7}$$

where *B* is the photographic basis (see Fig 3), $\delta p = defocu\sin g\_error\sqrt{2}$ is a parallax error, *n* is the number of images on which the point is portrayed.

From equation it is clear that it is necessary to find the defocusing error $\delta p$. The solution for the defocusing error will be the key to surveying distance finding.

To find the surveying distance, one must use the main principles of the geometric optic. The first step is the calculation of defocusing error. The defocusing error is defined through the aperture that is presented as a relationship [2,23]:

$$D = \frac{f}{N} \tag{8}$$

where *N* is well-known as f-number $N = \frac{f}{D}$.

The critical elements of the system object-lens-image are given in Fig 4.

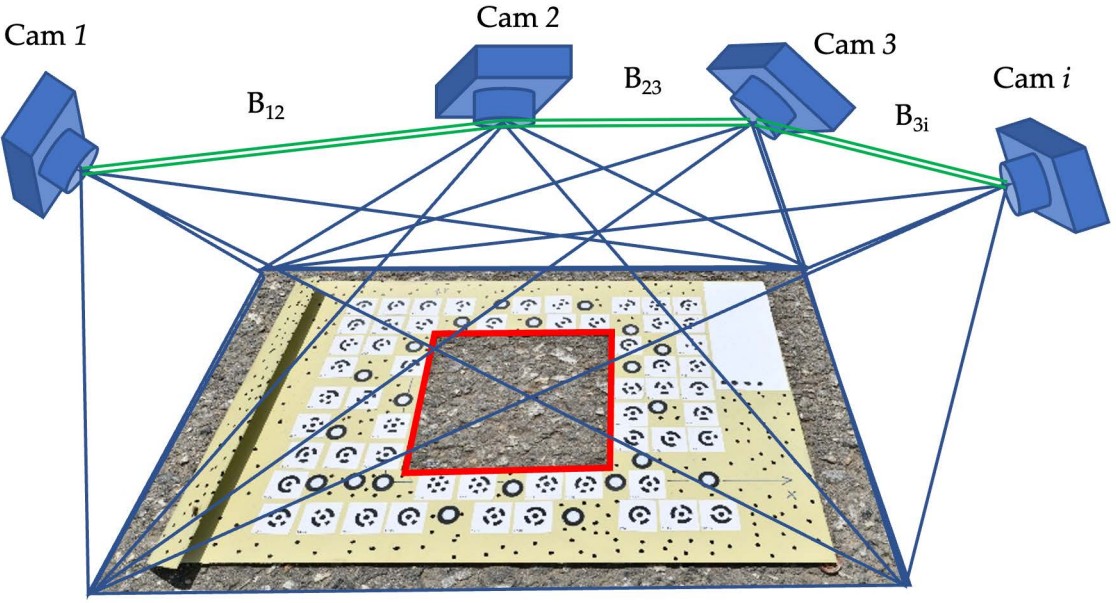

**Fig 3. Photogrammetric survey.**

From similar triangles in Fig 3 and after simple conversions, the defocusing error is:

$$\delta = \frac{f^2}{N}\left|\frac{(L-L')}{L'(L-f)}\right|$$

(9)

where $f$ is a focal distance, $L, L'$ are distances in object space.

For the analysis, the following values have been accepted: focal distance in a range from 5 mm up to 50 mm; surveying distance from 0.2 m to 1.1 m; f-numbers 1, 1.4, 2, 2.8, 4, 5.6, 8, 11, 16, 22; the detectable texture depth ± 5 mm. The first step is the image scale calculation. The results of this calculation are conveniently presented in the graph (Fig 5).

By the formula (9), it is possible to build the graph of the defocusing error changing (Fig 6). Which is a graphic summary of the defocusing error distribution depending on focal distance, surveying distance, and f-number.

To convert the values of the defocusing error to the errors of MTD determination, the expression (9) has been used. The results of this conversion are better presented in graphic form (see Figs 7, 8, 9 and 10). In these graphs, the different color schemes have been applied to distinguish the errors of MTD for different baseline lengths.

The red plane in the figures shows the standard error equals 0.223 mm, according to [3]. All values that are below the plane correspond to the allowable errors. The findings reveal a high dependency between the MTD accuracy due to defocusing error and the number of images at which the particular point is portrayed. Thus, there is no point in increasing the number of images over twenty since there is no significant accuracy improvement for more than mentioned images number.

From the graphs in Figs 7–10, one may infer that the longer baseline, the longer the surveying distance at which the necessary accuracy is provided. However, considering that the frame plate size is approximately 300 × 300 mm with a hole of 110 × 110 mm, the baseline cannot exceed 200–250 mm. The typical baseline length is about 100–150 mm. Therefore, the

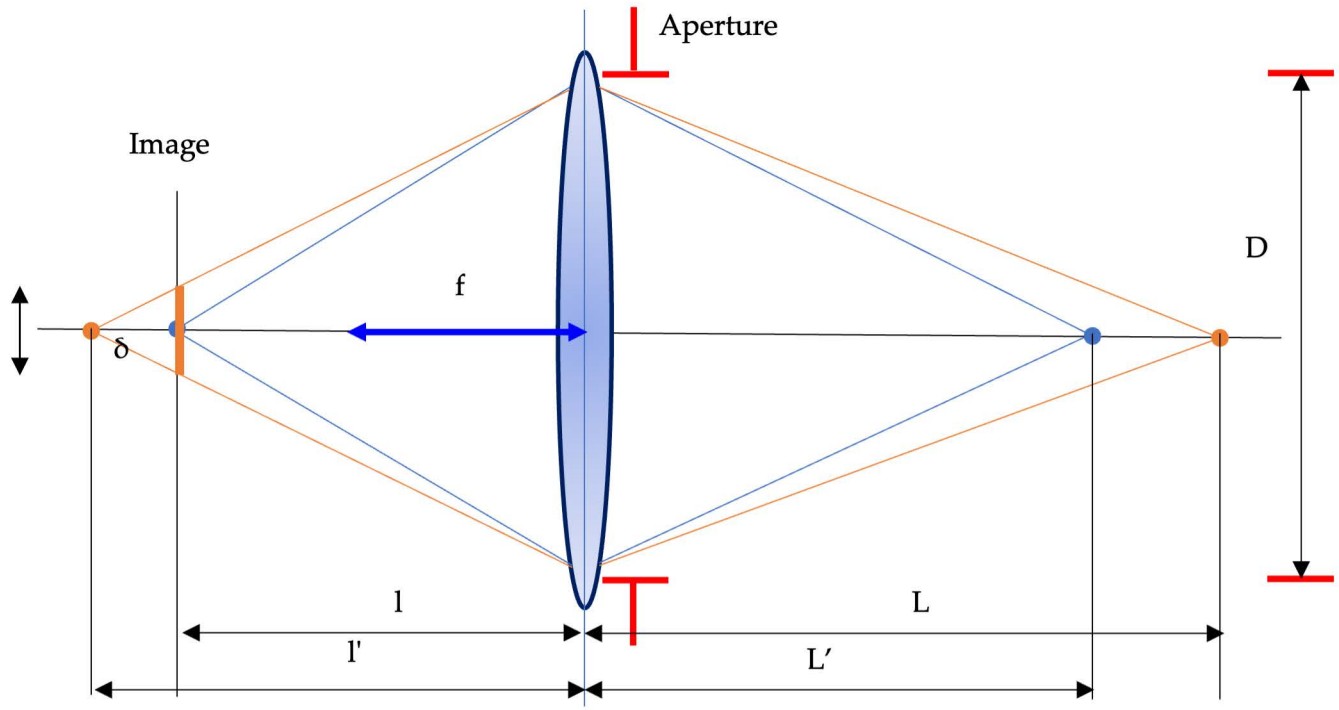

**Fig 4. The scheme of the defocusing error forming.**

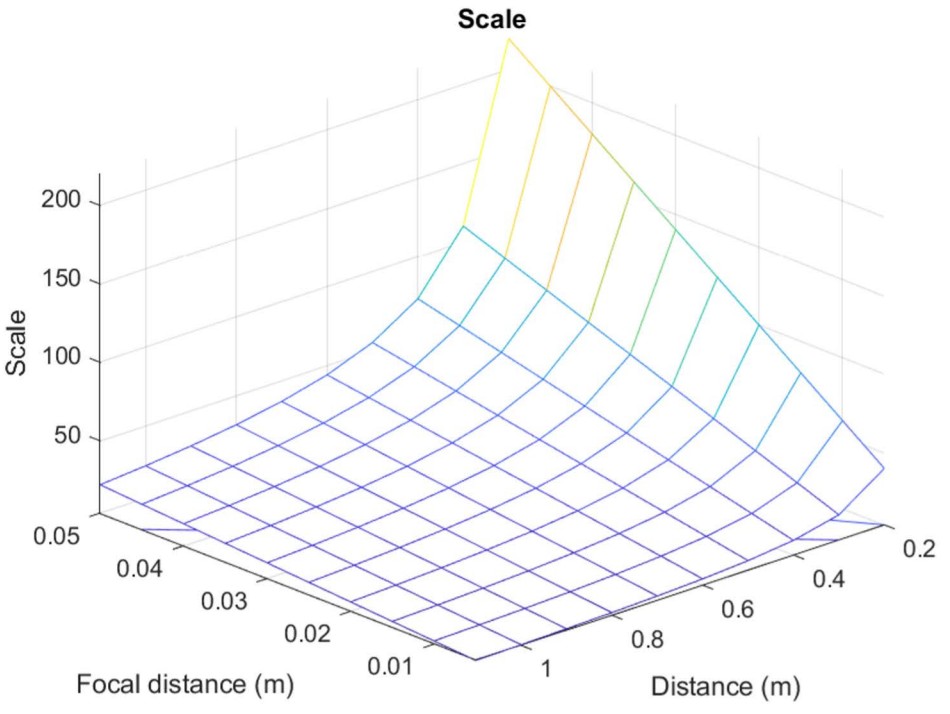

**Fig 5. Scale distribution.**

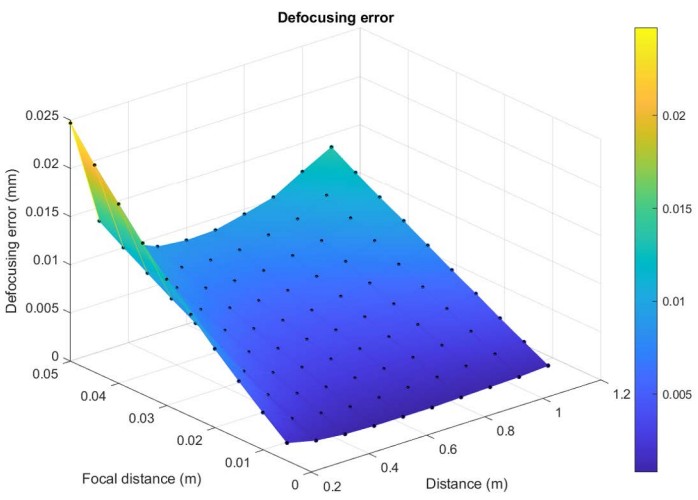

**Fig 6. Surface of the defocusing error distribution.**

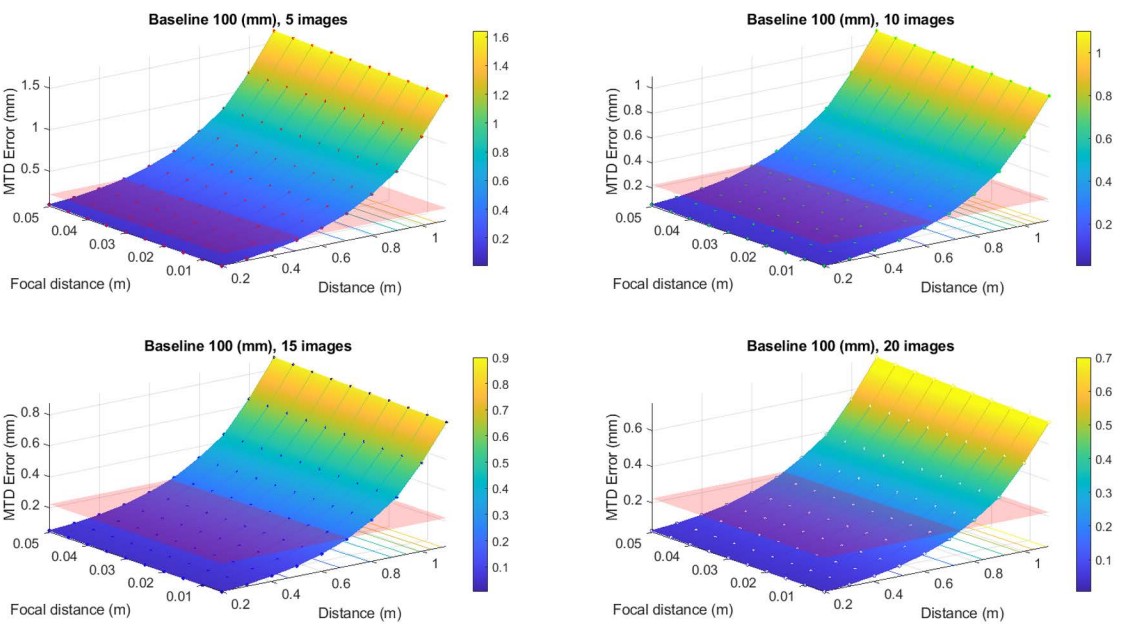

**Fig 7. Defocusing error for various image numbers and baseline 100 mm.**

optimal surveying distance is in the range of 0.3–0.6 m. However, it is evident that these range values for surveying distance are given for defocusing error. Then we may assign the necessary distance using the allowable value of the blur circle. The blur circle size δ refers to the depth of field. The depth of field describes the permissible difference in distances L2–L1 for which the blur circle does not exceed some allowable value (Fig 11).

The size of the blur circle can be calculated by the following expressions [3]:

$$\delta = \frac{f^2\left(L - L_1\right)}{NL_1\left(L - f\right)}, \delta = \frac{f^2\left(L_2 - L\right)}{NL_2\left(L - f\right)} \tag{10}$$

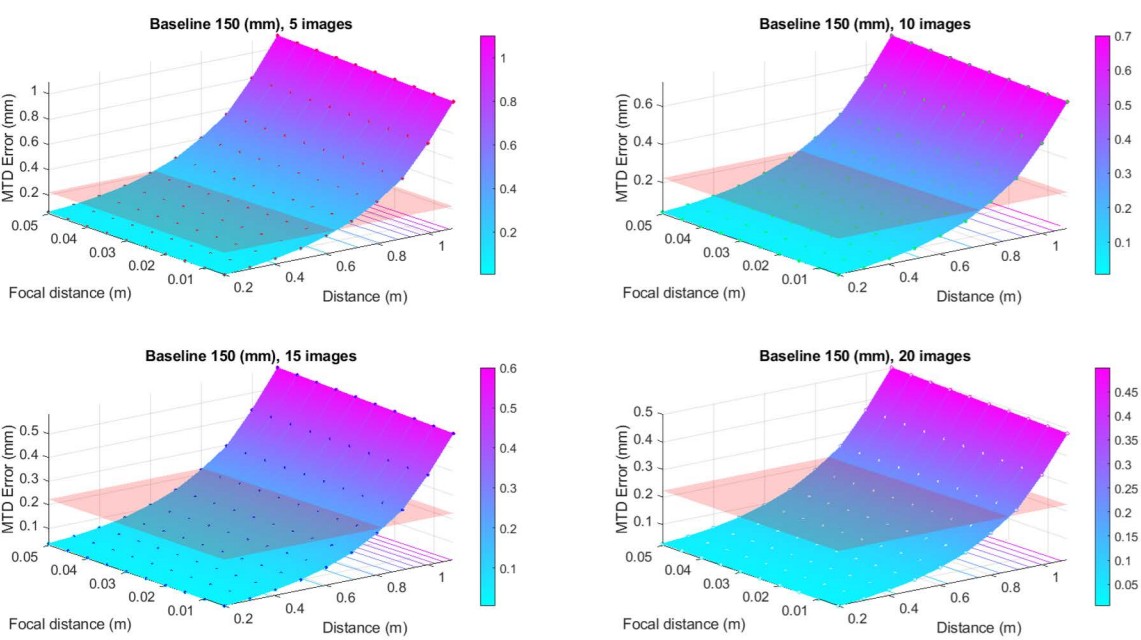

**Fig 8. Defocusing error for various image numbers and baseline 150 mm.**

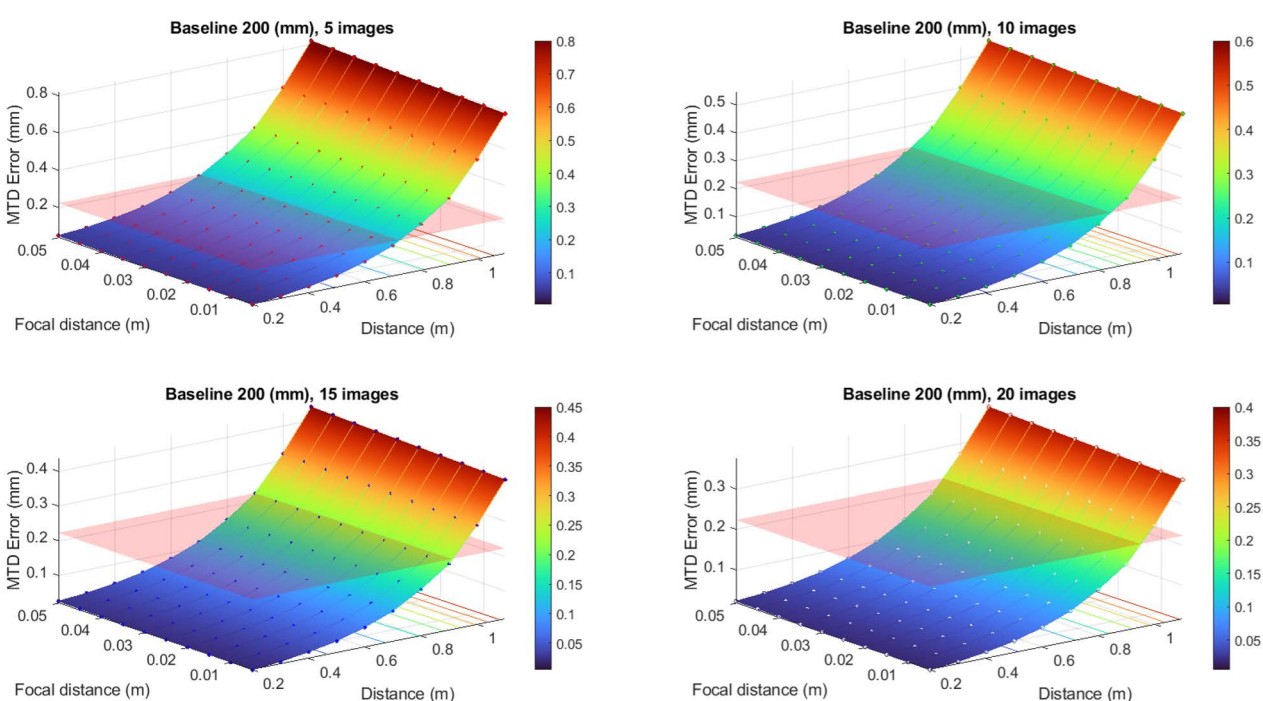

**Fig 9. Defocusing error for various image numbers and baseline 200 mm.**

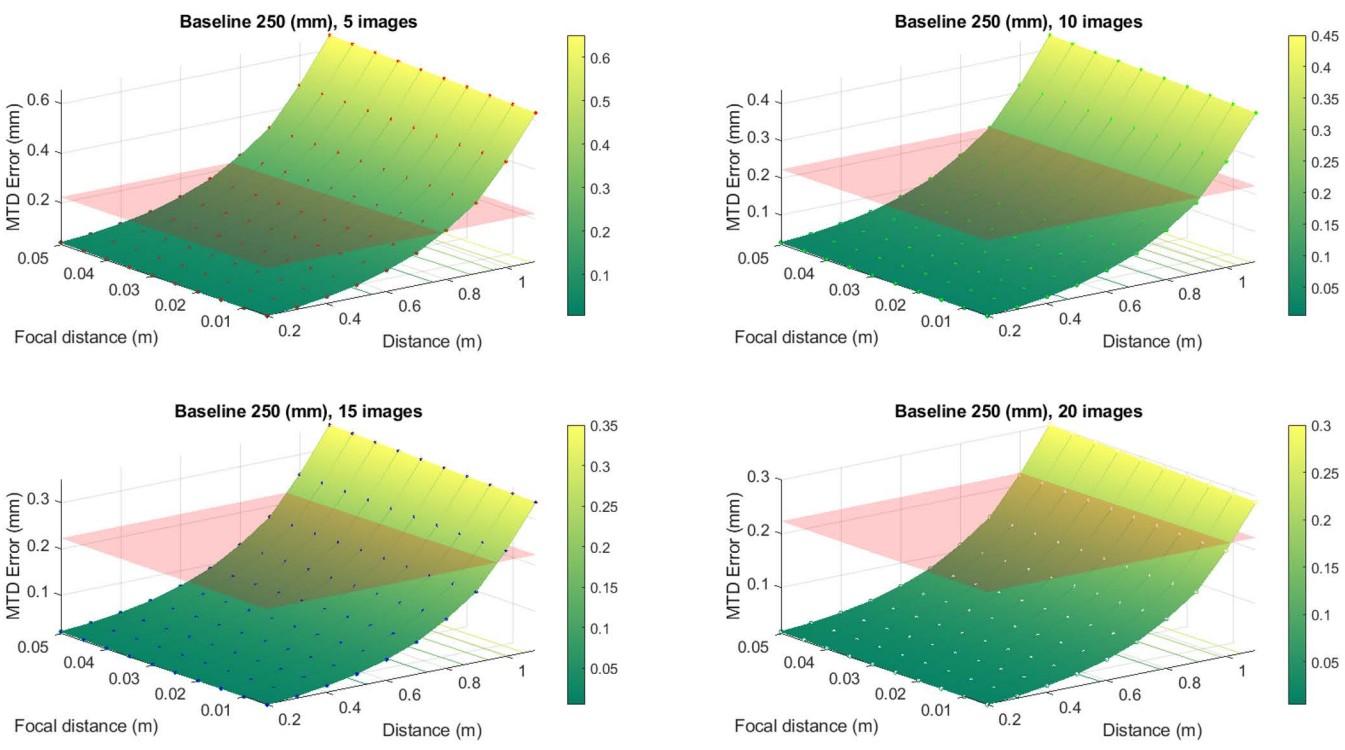

**Fig 10. Defocusing error for various image numbers and baseline 250 mm.**

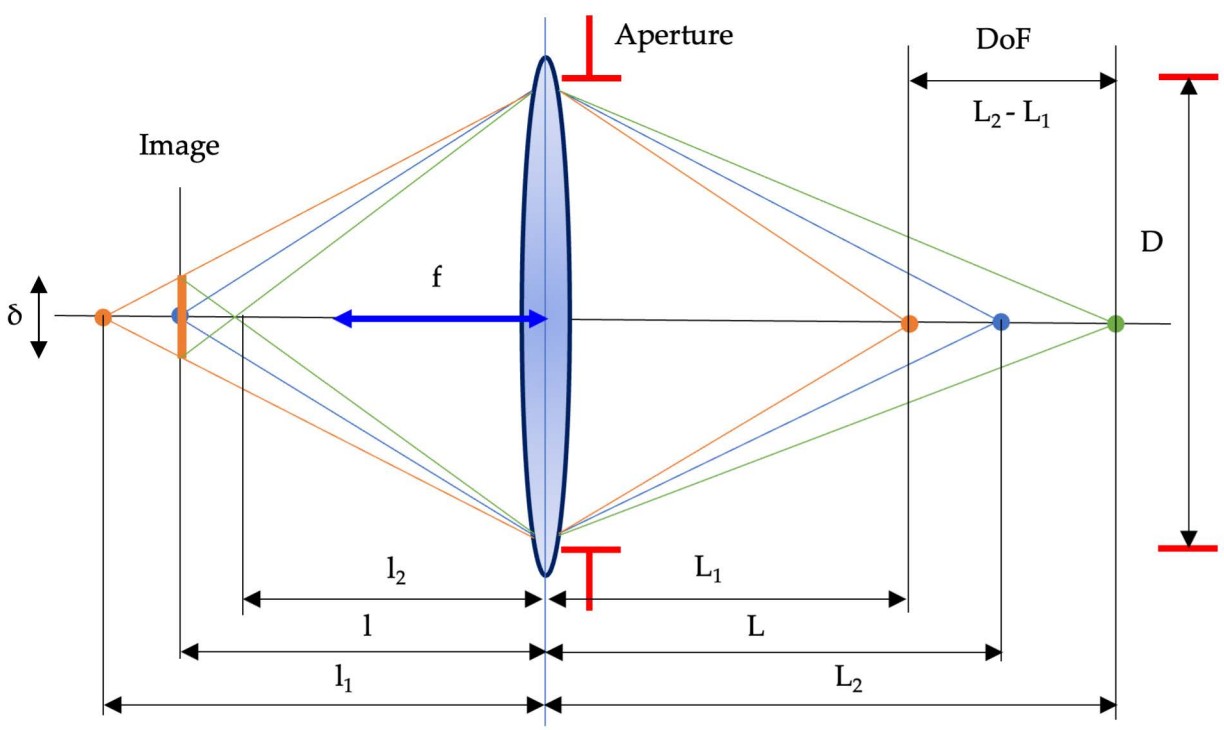

**Fig 11. The scheme for the depth of field calculation.**

From (10), one may find the depth of field

$$L_2 - L_1 = \frac{2Lf^2 \delta N (L - f)}{f^4 - \delta^2 N^2 (L - f)^2} \tag{11}$$

where all the designations are clear from the previous figures and expressions.

For precise measurements, one may suppose that the allowable size of the blur circle should not exceed the size of the pixel of the camera matrix. Thus, the blur circle's size has to range from 3 to 6 μm for typical consumer-grade cameras. The depth of field has been calculated for these two values and the given sets of f-numbers, focal distances, and surveying distances. In Figs 12 and 13, the surface of the depth of field distribution for a 3 μm blur circle is presented. The red plane in Fig 12 illustrates the typical range of the pavement's depth that corresponds to the image field of depth. All the values above this plane correspond to the allowable values of the blur circle. To grasp Fig 12 quickly, the top view of the surface is portrayed in Fig 13. Therefore, the surveying distance of 0.3 m can be considered very extreme (the same is valid for 0.9 m), and for such a blur circle size, the best surveying distance is in between 0.4 and 0.6 m.

The same calculations have been done for a 6 μm blur circle. In Figs 14 and 15, the surface of the depth of field distribution for a 6 μm blur circle is presented. As previously, the red plane in Fig 14 presents the range of the pavement's depth that corresponds to the image field of depth. All the values above this plane correspond to the allowable values of the blur circle. Fig 15 demonstrates the top view of the surface. Thus, the surveying distance is considerably larger than for 3 μm and for a 6 μm blur circle. The distance can be set at any length but

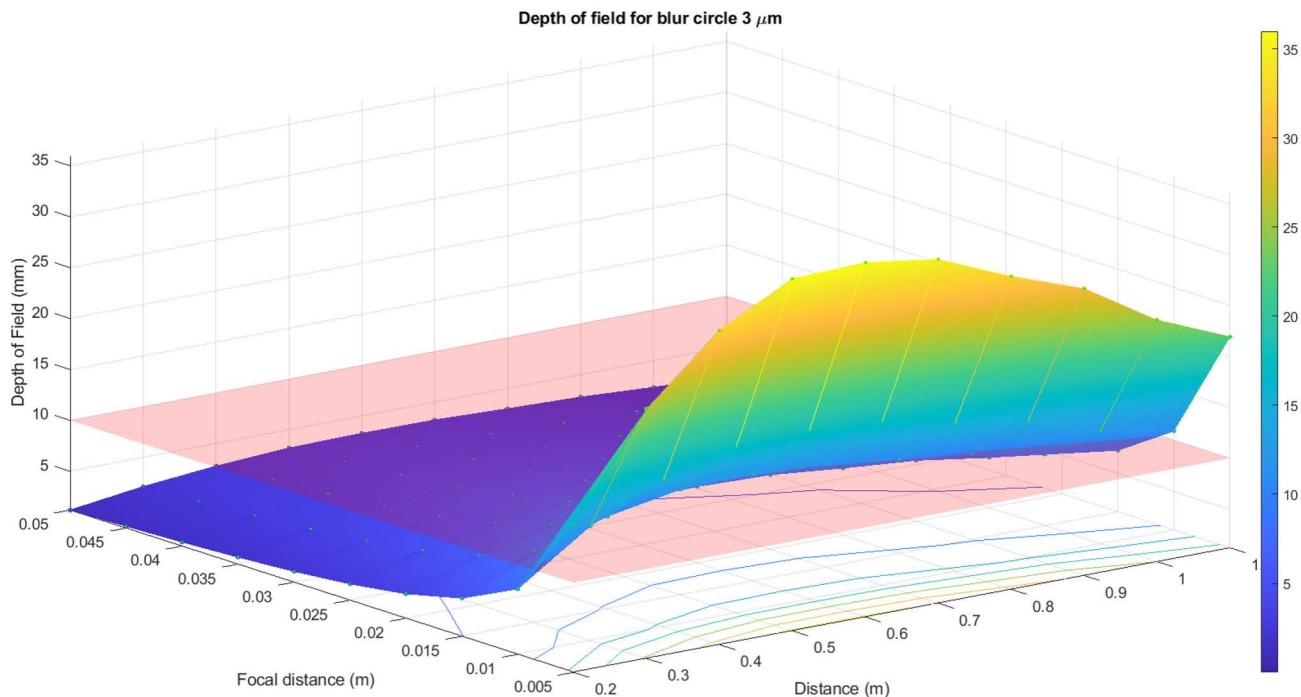

**Fig 12. Surface of the depth of field distribution for 3 μm blur circle.**

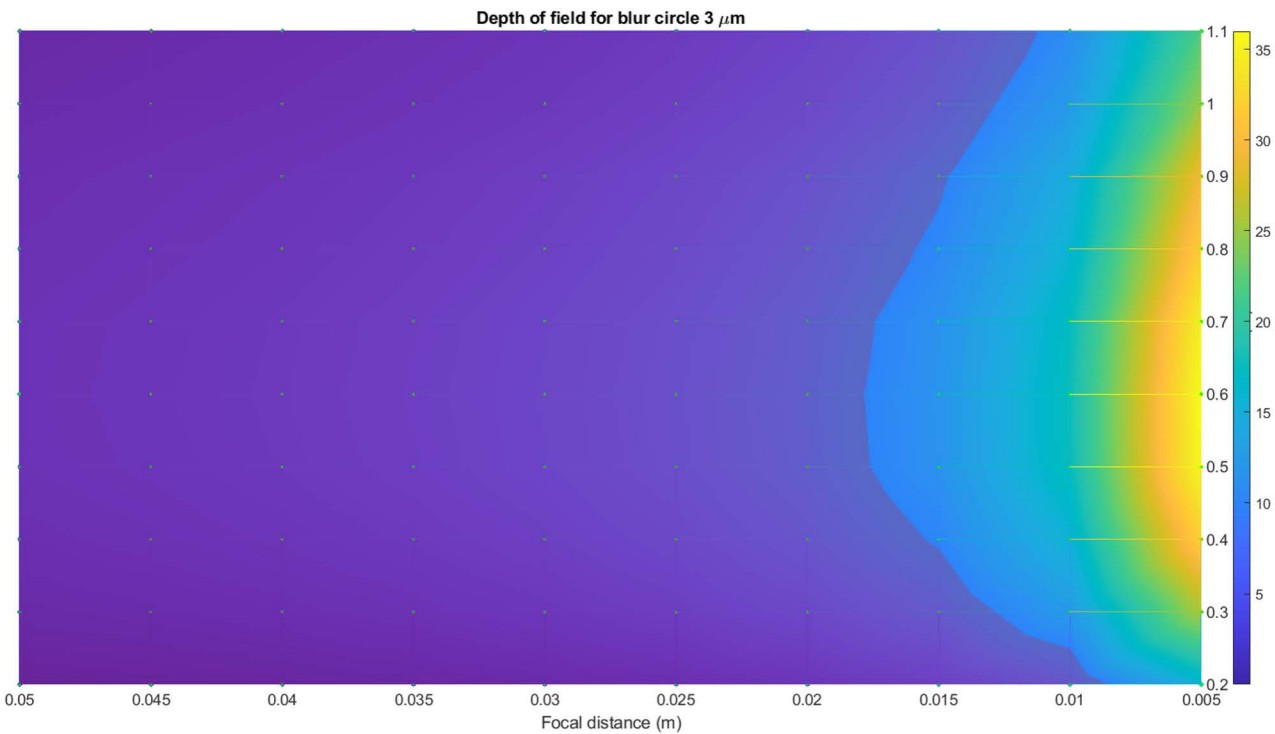

**Fig 13. Surface of the depth of field distribution for 3 μm blur circle top view.**

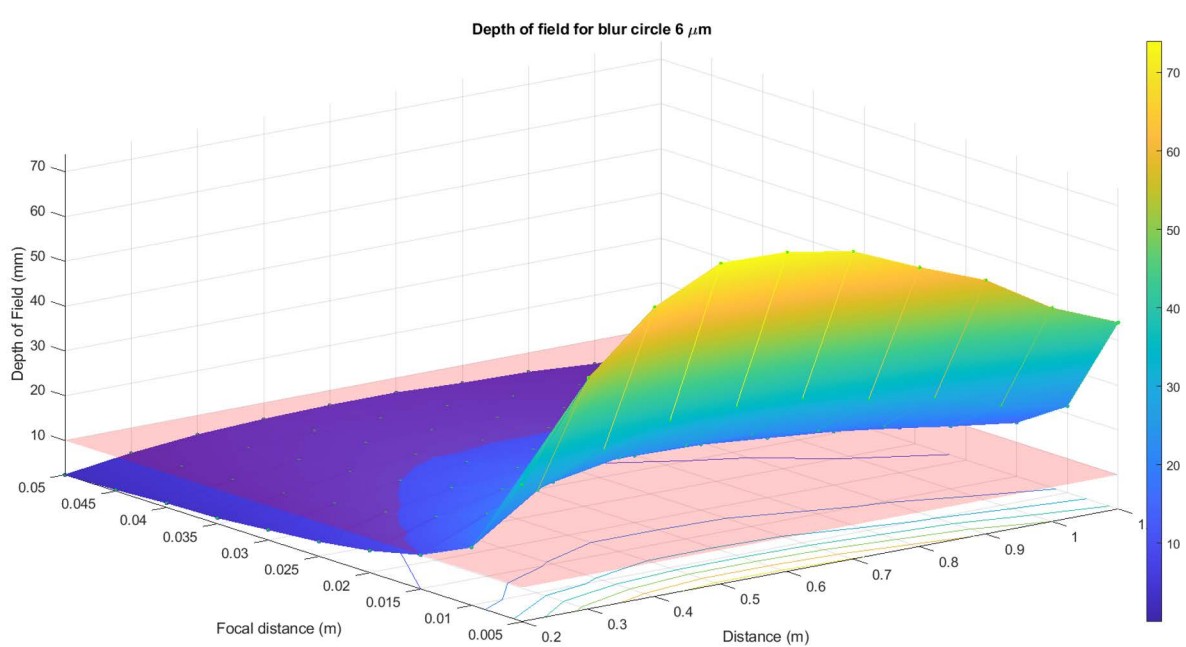

**Fig 14. Surface of the depth of field distribution for 6 μm blur circle.**

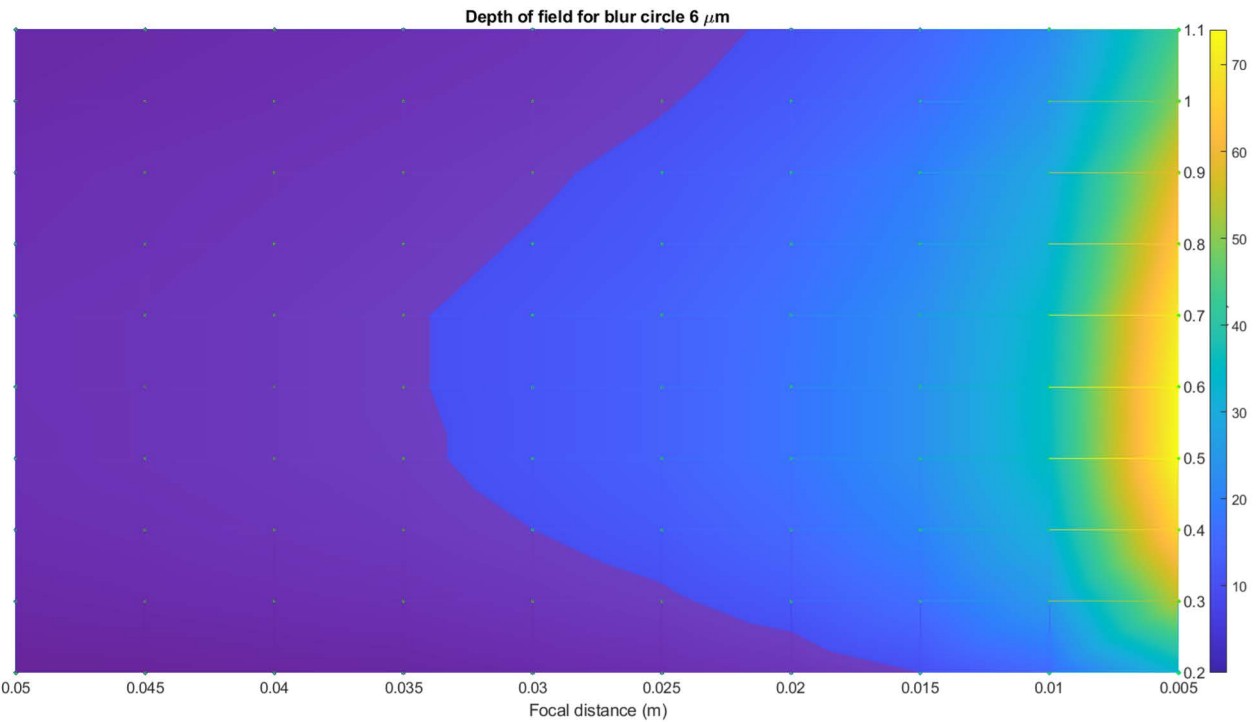

**Fig 15. Surface of the depth of field distribution for 6 μm blur circle top view.**

keeping in mind the appropriate values of f-number. For both cases, the same optical principle is valid: the smaller f-number, the shallow depth of field, and vice versa.

Next step is to examine the possibility of using hyperfocal distance. Instead of calculating the depth of field, one may calculate the closest distance at which the lens is focused at infinity, and all the objects are portrayed in an image with allowable blur circle size (or, in other words, with the permissible level of sharpness, Fig 16). This distance is called hyperfocal and can be found by [2]

$$L_h = \frac{f^2}{\delta N} + f \qquad (12)$$

where all the designations are clear from the previous figures and expressions (Fig 16).

The graphs of hyperfocal distances for aperture D (8) equal to one are shown in Fig 17.

The shorter hyperfocal distance equals 0.8 m for a blur circle of 6 μm. The results show the useless of hyperfocal distance for road pavement friction detection since the hyperfocal distances are too long. The longer distances lead to small-scale images and downgrade the accuracy. Summing up this study, surveying distances of 0.4 – 0.6 m is recommended to ensure the small blur circle size and allowable defocusing error.

## Data description

The places for the test data collection were chosen according to the recommendations suggested in the standard [22]. For each test spot, the MTD was determined by two approaches: SPM and photogrammetric. The photogrammetric determination is as follows (Fig 18):

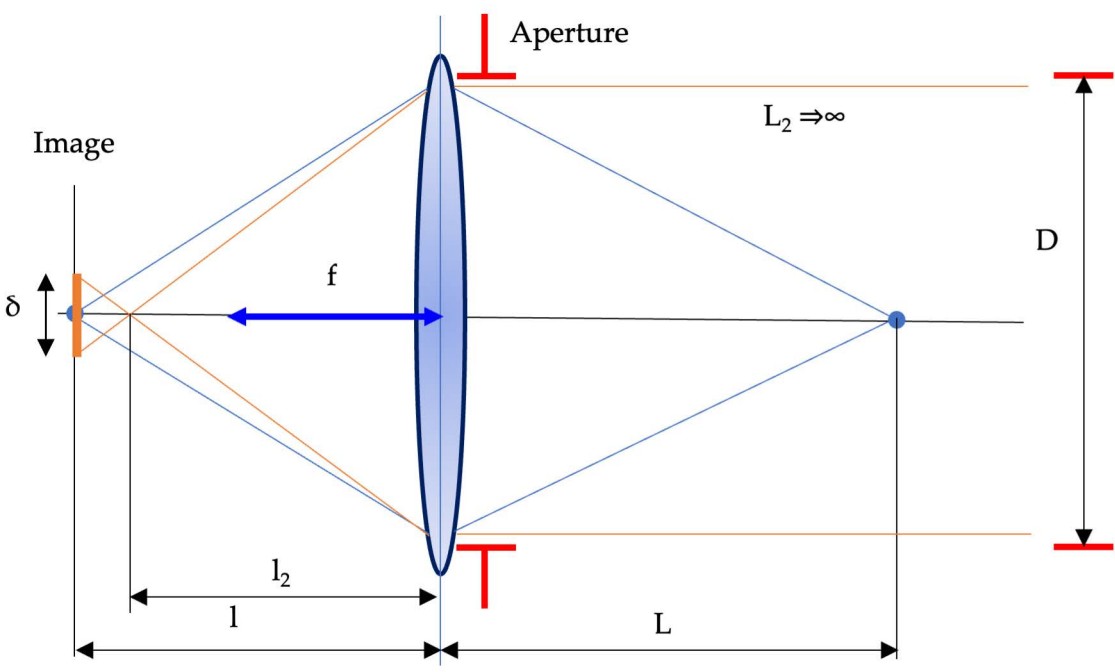

**Fig 16. The scheme for the hyperfocal distance calculation.**

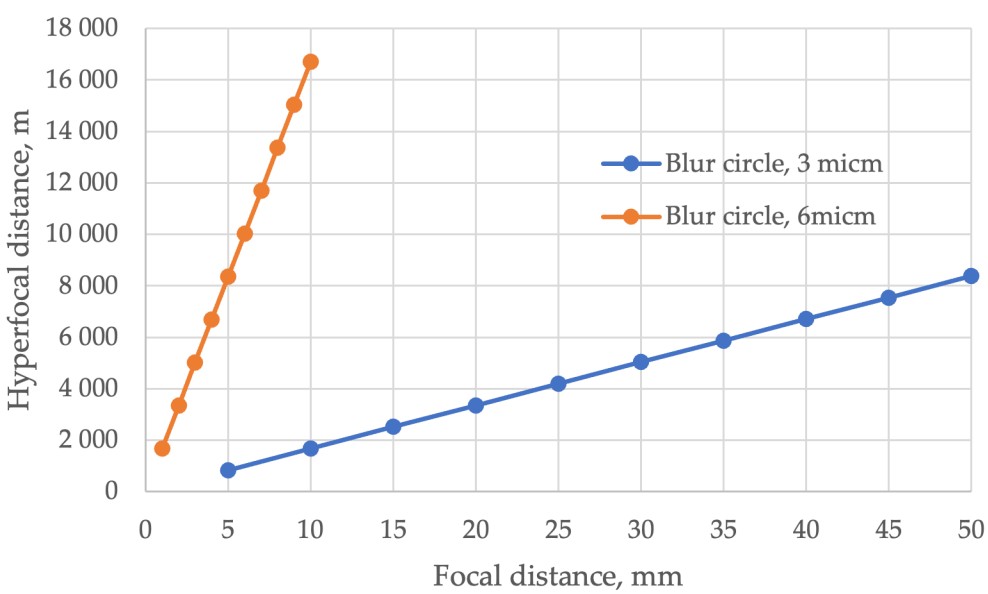

**Fig 17. Hyperfocal distances.**

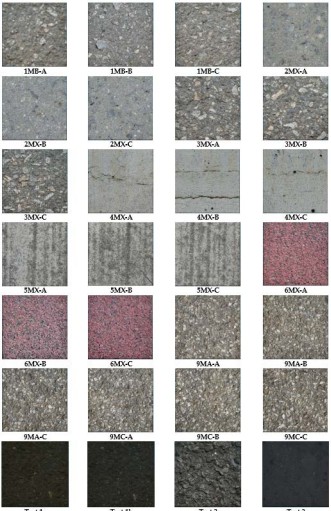

**Fig 18. Generated point clouds for various pavement types.**

## Simulation scheme and algorythms

In the introduction section, it has been pointed out that a standardized approach based on simple selection of three points with maximum height cannot ensure a reliable solution due to evident reasons. That is why it was decided to develop a refined approach to MTD calculation. The new modified approach is based on the study of the different calculation schemes (Fig 19). The study of these schemes will allow us to determine the new best approach for MTD calculation.

The core elements of the study are the approach of the reference plane determination, the number of points used for the reference plane draw, the partitioning scheme according to which the initial test patch is split, and a masking region that constrains the area for point search.

There are three methods of reference plane drawing. The simplest one is the horizontal plane. The only condition is selecting the point with maximum height inside the test patch and drawing the plane through this point. The plane equation in inexplicit form is [3]:

$$AX_i + BY_i + CZ_i + D = 0 \tag{13}$$

where $A, B, C, D$ are plane coefficients. To reduce the computational complexity, eq. 13 was rearranged as follows:

$$\left(\frac{A}{D}\right)X + \left(\frac{B}{D}\right)Y + \left(\frac{C}{D}\right)Z = -1 \tag{14}$$

Therefore, for the horizontal plane, we have the following:

$$0X + 0Y + Z = h_{max}. \tag{15}$$

Equation (15) has three unknowns; therefore, three points define the exact plane. Otherwise, the solution is found using the singular value decomposition method that provides best-fit

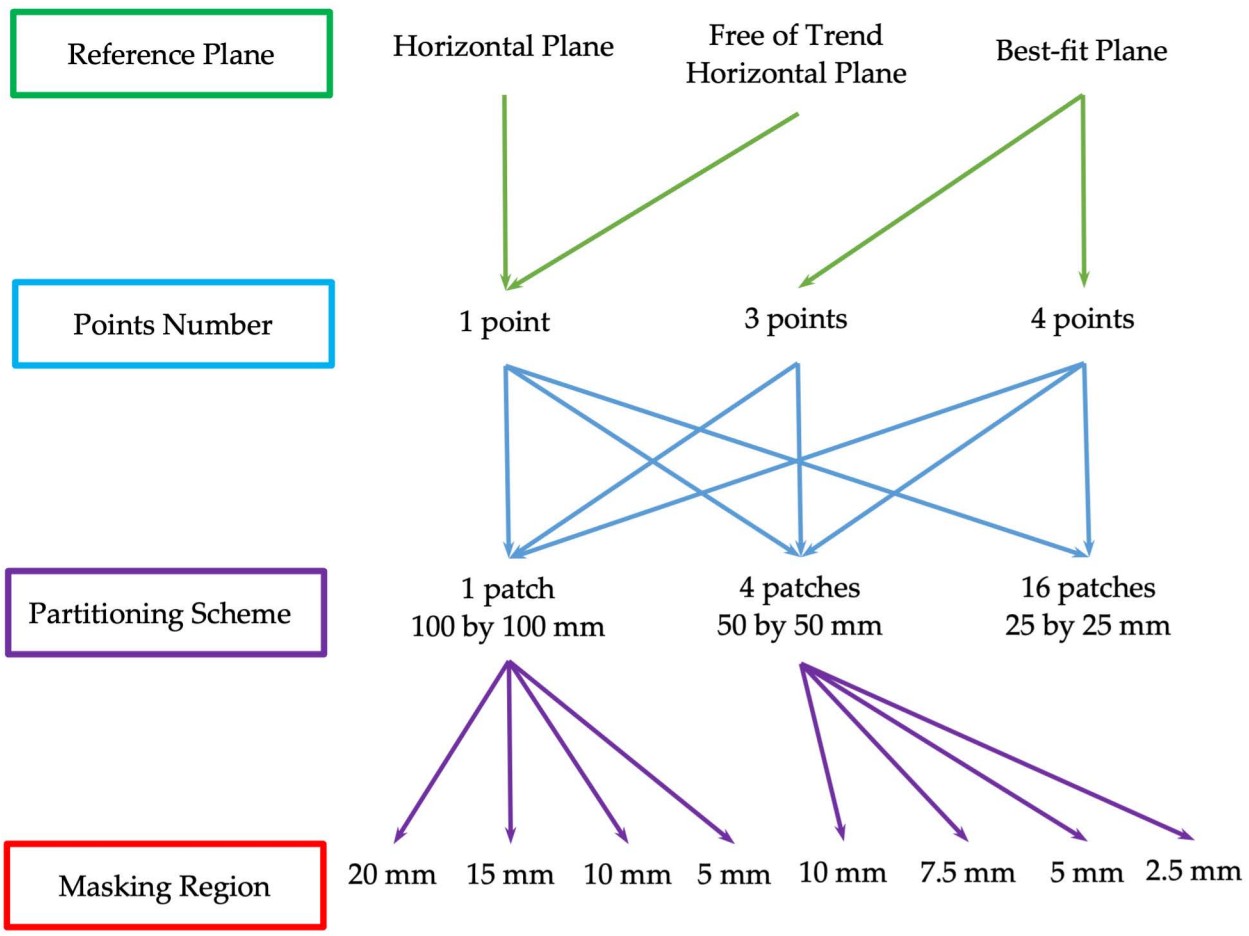

**Fig 19. Simulation scheme.**

plane estimation if we have more than three points. The second method is the free-of-trend method. The idea is to approximate by plane equation (14) the test patch using the singular value decomposition method. This plane is considered a trend plane, and the test patch is being transformed according to this trend plane. For the transformed plane, searching for the maximum height point is accomplished. The last method is the best-fit plane. This method uses preselected points to draw the reference plane throughout these points using a least squares condition.

Points' numbers may vary depending on the reference plane draw method. A single point is used for the horizontal plane method (15). Three or four points are used for the best-fit plane method.

The partitioning scheme is one of the primary suggestions. Different methods of the reference plane and the number of points were applied for test patches of various sizes. The simplest case is the test patch of full size ($100 \times 100$ mm). To reduce the impact of the local fluctuations of the whole test patch, it was suggested to split it into four smaller patches ($50 \times 50$ mm) and apply the mentioned methods to all these patches separately. Then MTD (6) is calculated as an average value of these particular values MTDi.

$$MTD = \sum_{1}^{4} MTD_i / 4 \qquad (16)$$

Splitting the initial test patch into sixteen patches was done to estimate the effectiveness of the partitioning strategy. As previously, the MTD value is calculated as an average value. The partitioning of the initial test patch is given in Fig 20.

The second important suggestion is the masking region. The idea is based on the scheme that allows overcoming the issue of the extreme proximity of points with maximum heights. That case has been discussed in the introduction. It was offered to overlay the specific mask to cut down this effect. Inside the mask, only one point can be chosen. For different partitioning schemes, the different mask sizes were suggested and studied.

## IV. Results

According to the scheme in Fig 21, twenty-three different approaches were studied. The first three cases were considered for the simple horizontal plane with varying partitioning schemes (1 patch, 4 patches, 16 patches). These approaches are called "Horizontal plane max point 1 patch", "Horizontal plane max point 2 × 2 4 patches", and "Horizontal plane max point 4 × 4 16 patches". The result for the test patch 1MB-A is presented in Fig 21.

The surface volumes below the horizontal planes were calculated for all data sets with different partitioning schemes, and MTD values were determined.

In the next step, the horizontal planes for the same data but after the trend exclusion have been calculated. This task is based on the least square approximation of the test patch using equation (15). Fig 22 shows the data set 1MB-B and its trend plane. The data set is mapped onto the trend plane. Consequently, we obtain a test patch free of trend. Such a simulation has been accomplished for two partitioning schemes (1 patch and 4 patches). These approaches are called "Horizontal plane max point 1 patch FoT" and "Horizontal plane max point 2 × 2 4 patches FoT". The surface volumes below the horizontal planes for free-of-trend data sets have been calculated, and MTD values were determined.

The rest of the simulations were done for three or four points with maximum heights. Earlier, we pointed out that the standardized approach does not work for the whole test patch. It was recommended to perform simulations for four patches. The key concept is to divide the test patch into four equal patches. Then, inside each of the patches, we determine the single point with a maximum height. These points are used to derive the best-fit plane through three or four points. These approaches are called "Best-fit plane 3 max points 4 patches" and "Best-fit plane 4 max points 4 patches". In Fig 23, the best-fit planes via three and four points for the 2MX-A data set are given.

The images demonstrate that the planes are highly inclined due to local roughness and do not describe the existing pavement surface. It was decided to go down in partitioning and try to divide the test patch into four and sixteen patches. An example of the best-fit plane construction for the 2MX-B data set is presented in Fig 24. There are four patches inside of each four points with maximum heights selected, and the plane throughout them is drawn. The same case, but for three points, is portrayed in Fig 25.

It is evident that despite the more detailed partitioning, the suggested approach did not work well since locally particular planes significantly deviate from the actual surface. So, we tried to calculate MTD using partitioning into sixteen patches. An example of the best-fit plane construction for the 1MB-A data set is presented in Fig 26. There are sixteen patches. Inside each patch, four points with maximum heights are selected, and the plane throughout them is drawn. The same case for three points is given in Fig 27.

1 patch 100 by 100 mm

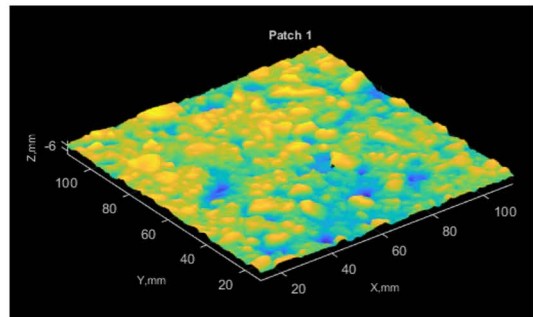

4 patches 50 by 50 mm

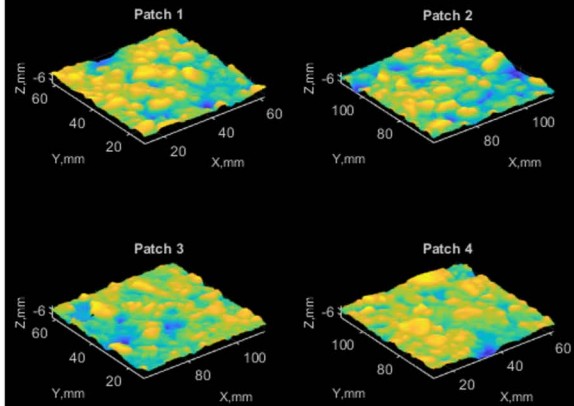

16 patches 25 by 25 mm

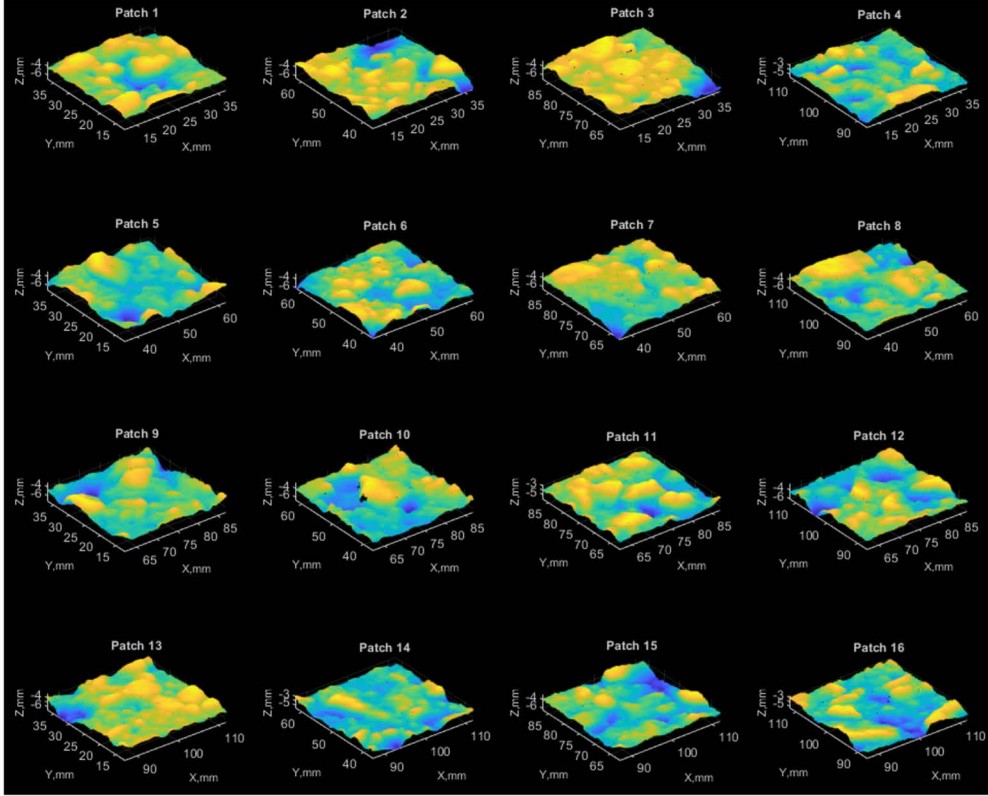

**Fig 20. Partitioning results for the initial test patch.**

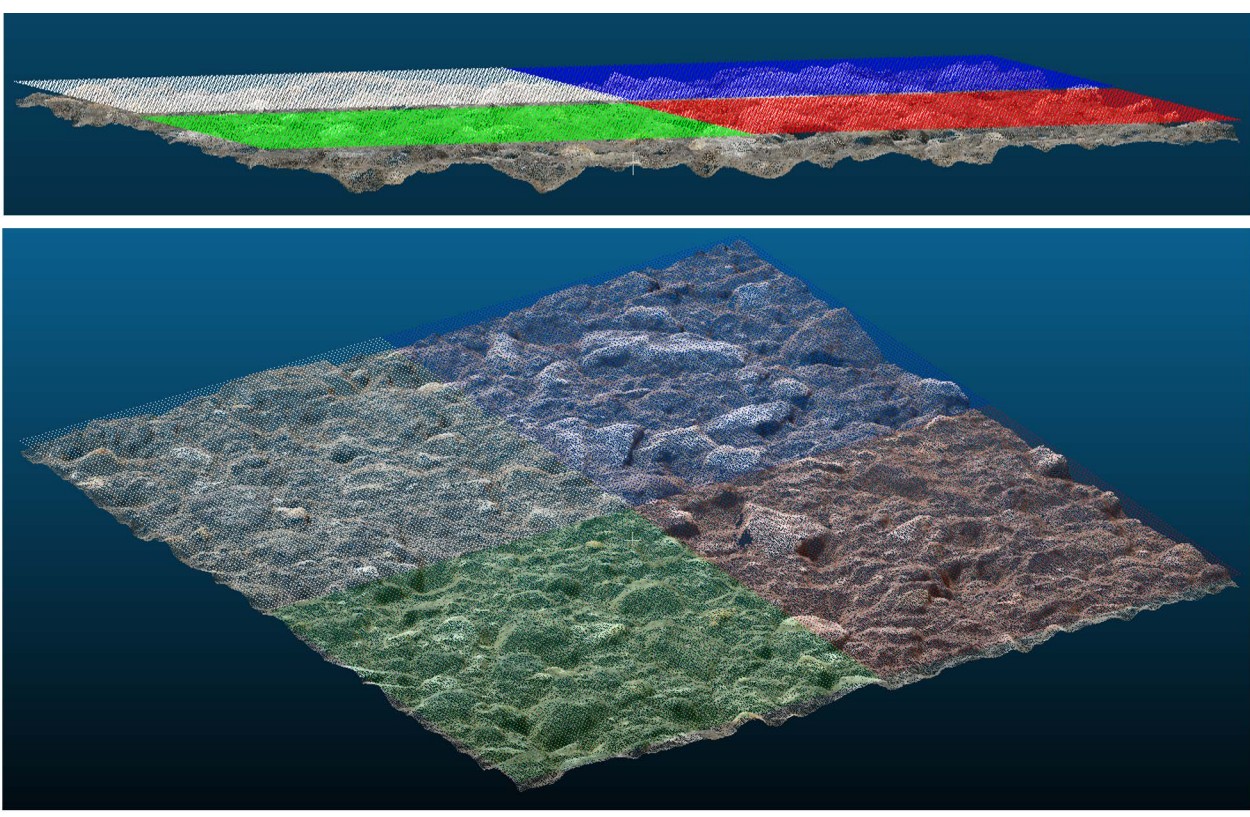

**Fig 21. A case study of horizontal plane 2 × 2, patch size 50 mm.**

It was found that there is still a case when this approach does not provide reliable results. Therefore, it was decided that there was no point in such an approach.

The primary efforts were focused on the development and study of the approach that is based on the masking region. Using the whole test patch, the developed algorithm embarks on searching for the point with the maximum height. At once the point is found, the masking rectangular region is assigned around this point. The second point is being searched for the test patch, excluding the masked area. After determining the second point, the masking rectangular region around this point is assigned again. Thus, we define three or four points emplaced from each other at a distance that equals the mask size. Sixteen masking cases have been studied in total.

The studied approaches were called respectively for four points "Best-fit plane 4 points 1 patch mask size 5 mm", "Best-fit plane 4 points 1 patch mask size 10 mm", "Best-fit plane 4 points 1 patch mask size 15 mm", "Best-fit plane 4 points 1 patch mask size 20 mm", "Best-fit plane 4 points 2 × 2 4 patches mask size 2.5 mm", "Best-fit plane 4 points 2 × 2 4 patches mask size 5 mm", "Best-fit plane 4 points 2 × 2 4 patches mask size 7.5 mm", "Best-fit plane 4 points 2 × 2 4 patches mask size 10 mm", and the same titles but for three points.

As a case study, the results for the best-fit plane of four points with mask size 5, 10, 15, and 20 mm are presented in Fig 28 (test patch 6MX-C).

Another example is given in Fig 29. This figure demonstrates the case of the plane thru three points for four patches with a mask size of 2.5 mm.

The results for different MTD calculation approaches in table form are not representative. That is why the graphic view is the best way to present the calculation results. The whole

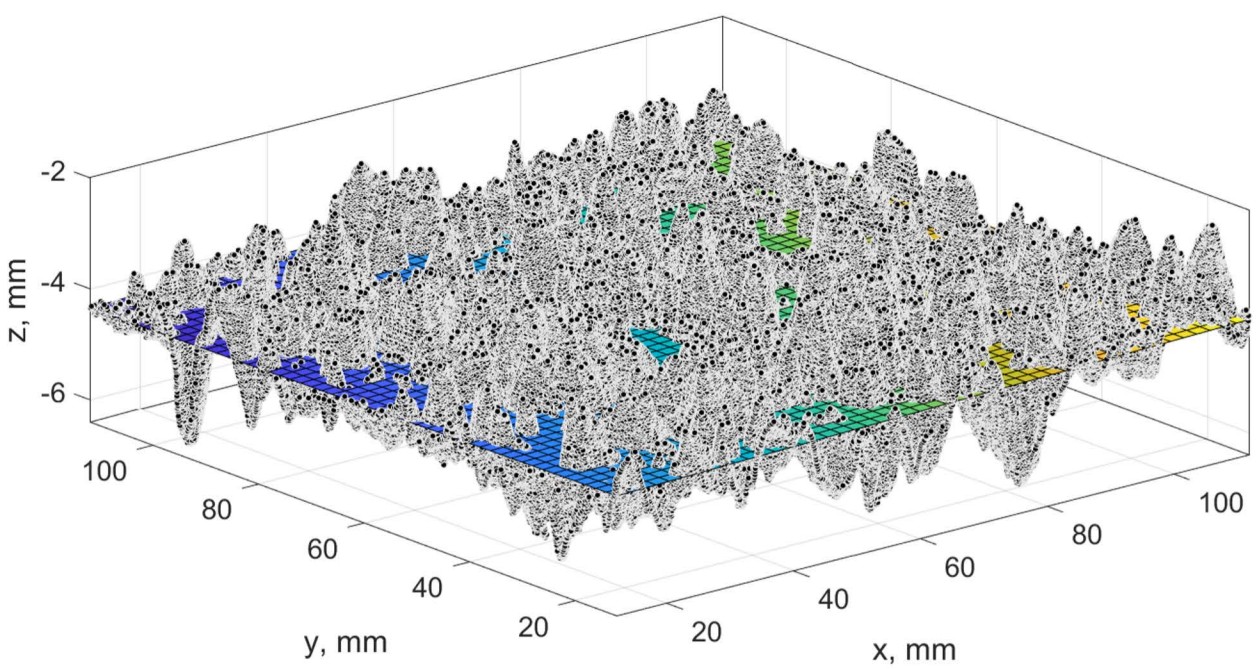

**Fig 22. Trend plane for a test patch.**

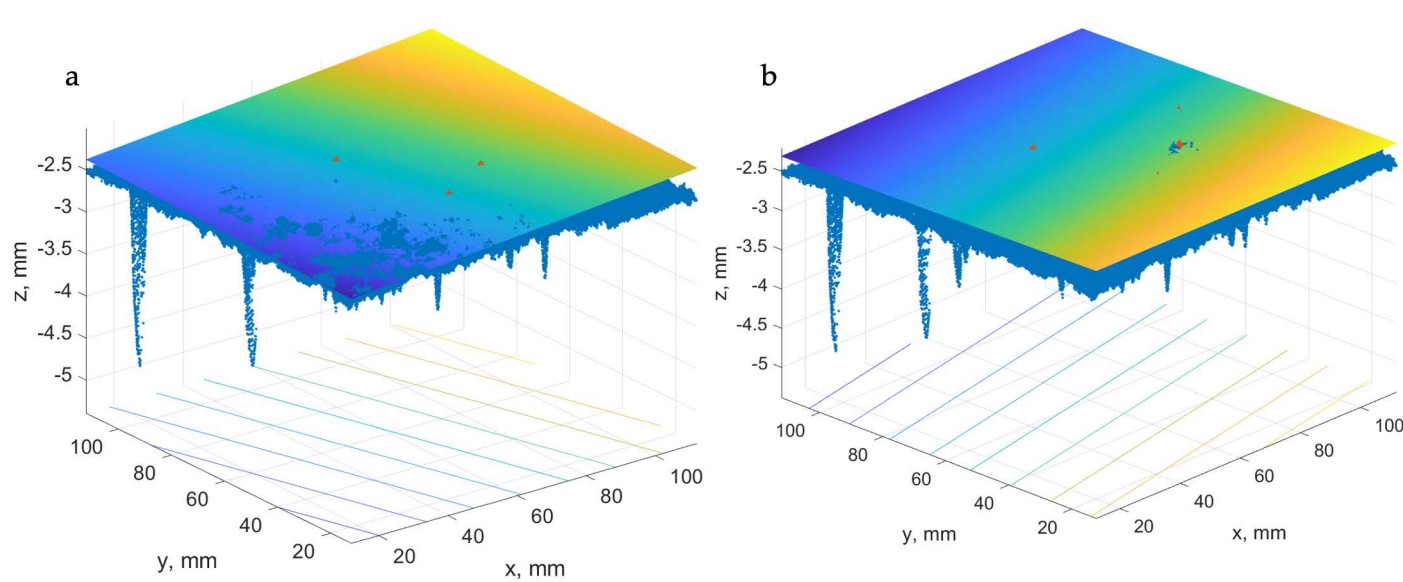

**Fig 23. A case study of the best-fit plane for** a) three points, b) four points.

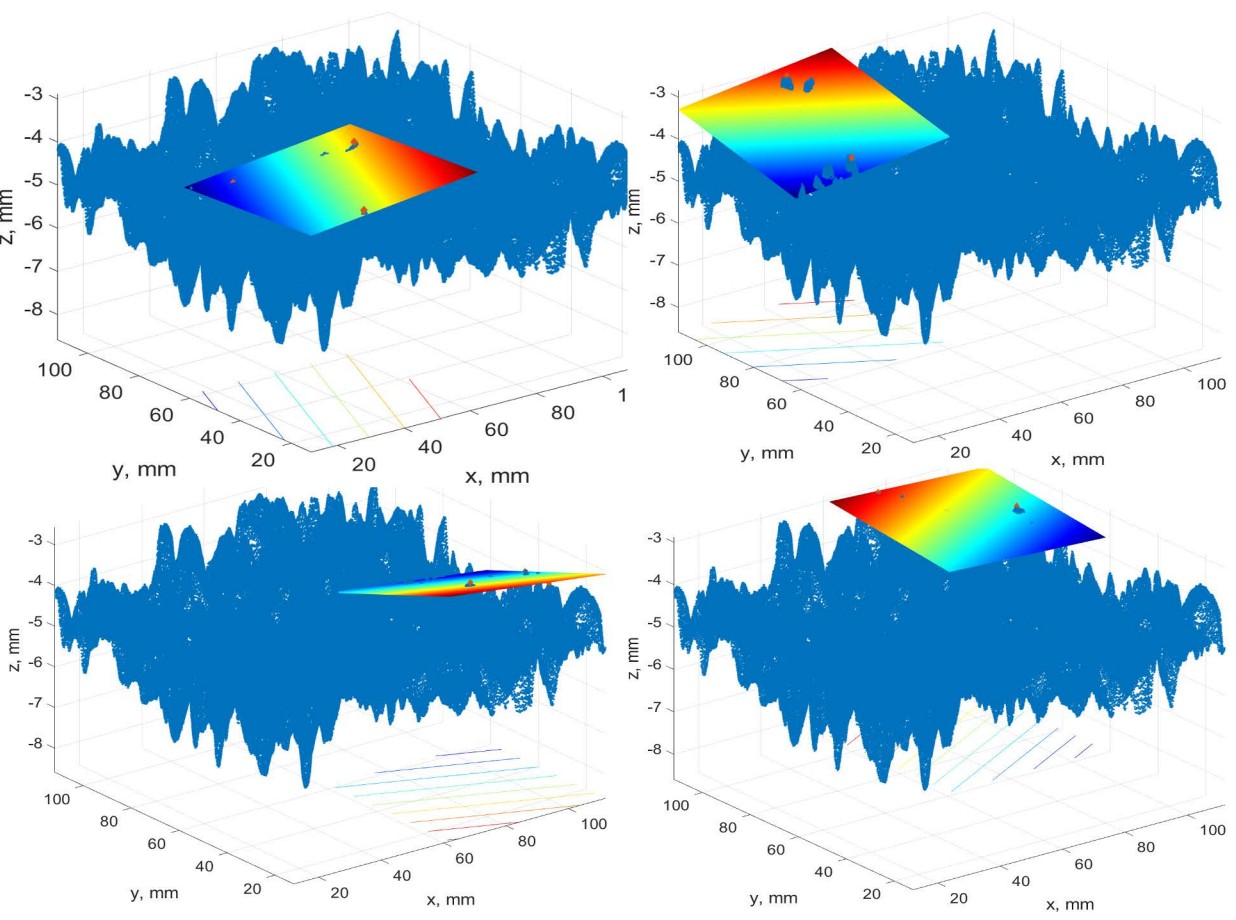

**Fig 24. A case study best-fit plane of 4 patches for four points.**

picture of the MTD values distribution can be grasped throughout the heatmap of their values (Fig 30), where the sand patch method is the reference one. The results of this method are considered without errors. In Fig 31, MTD values for different test sites are given.

Therefore, for analysis, one may compare the results of different approaches with the sand patch method. The comparison has been made using correlation analysis between each approach and the sand patch method. Various statistical measures have been used for the qualitative comparison, namely, mean square error, mean absolute error, mean absolute percentage error, and maximum error. The results of these measures' calculations are represented in Fig 32.

The final estimation was suggested to determine the best approach. The best estimation must have the smallest value among others. All statistical measures were normalized using expressions:

$$normR2_i = \frac{\frac{1}{R2_i}}{\sum_1^{23}\frac{1}{R2_i}}, norm\sigma_i = \frac{\sigma_i}{\sum_1^{23}\sigma_i}, normm_i = \frac{m_i}{\sum_1^{23}m_i}, normmp_i == \frac{mp_i}{\sum_1^{23}mp_i},$$

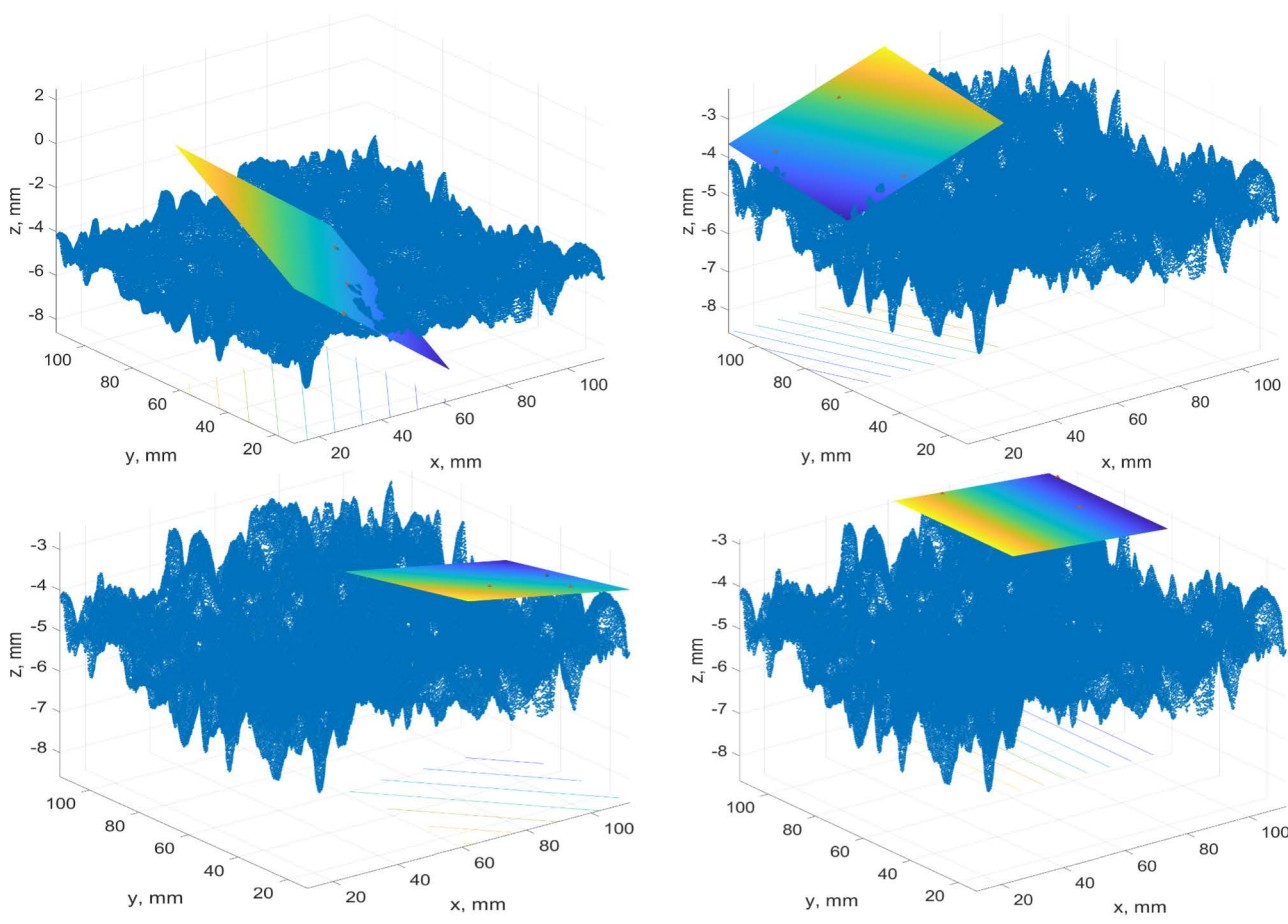

**Fig 25. A case study best-fit plane of 4 patches for three points.**

$$norm\delta_i = \frac{\delta_i}{\sum_1^{23}\delta_i}, norm\Delta_i = \frac{\Delta_i}{\sum_1^{23}\Delta_i}, normr_i = \frac{\frac{1}{r_i}}{\sum_1^{23}\frac{1}{r_i}}$$

The final estimation is obtained as:

$$E_i = normR2_i + norm\sigma_i + normm_i + normmp_i + norm\delta_i + norm\Delta_i + normr_i$$

The values from Fig 32 are used for the discussion of different approaches.

## V. Discussion

Investigating the road surface using photogrammetry seems to be a very promising alternative to other types of pavement friction measurements (e.g., the sand patch method). A great advantage is the possibility to see the investigated surface in greater detail, which can be projected retrospectively. At the same time, it is a new, modern way to assess pavement materials non-destructively. Complications can arise during the actual processing, as proper surface preparation and the quality of the photography are essential and directly dependent on the desired results.

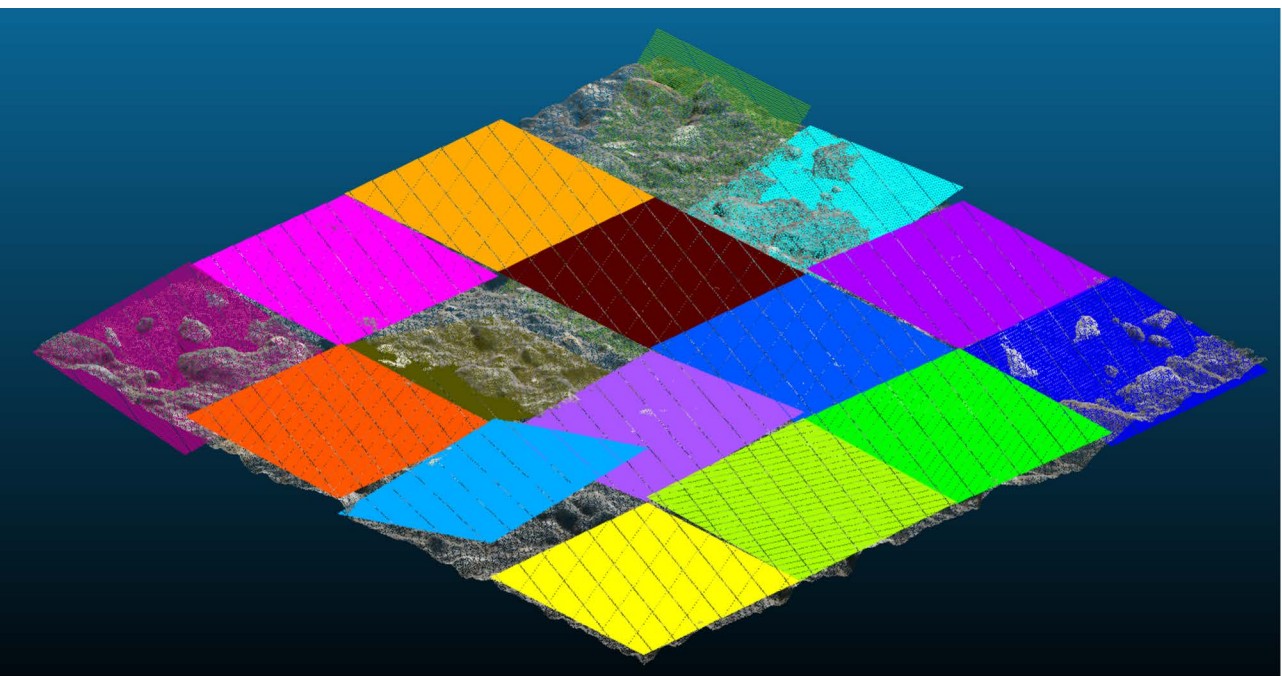

**Fig 26. A case study best-fit plane of 16 patches for four points.**

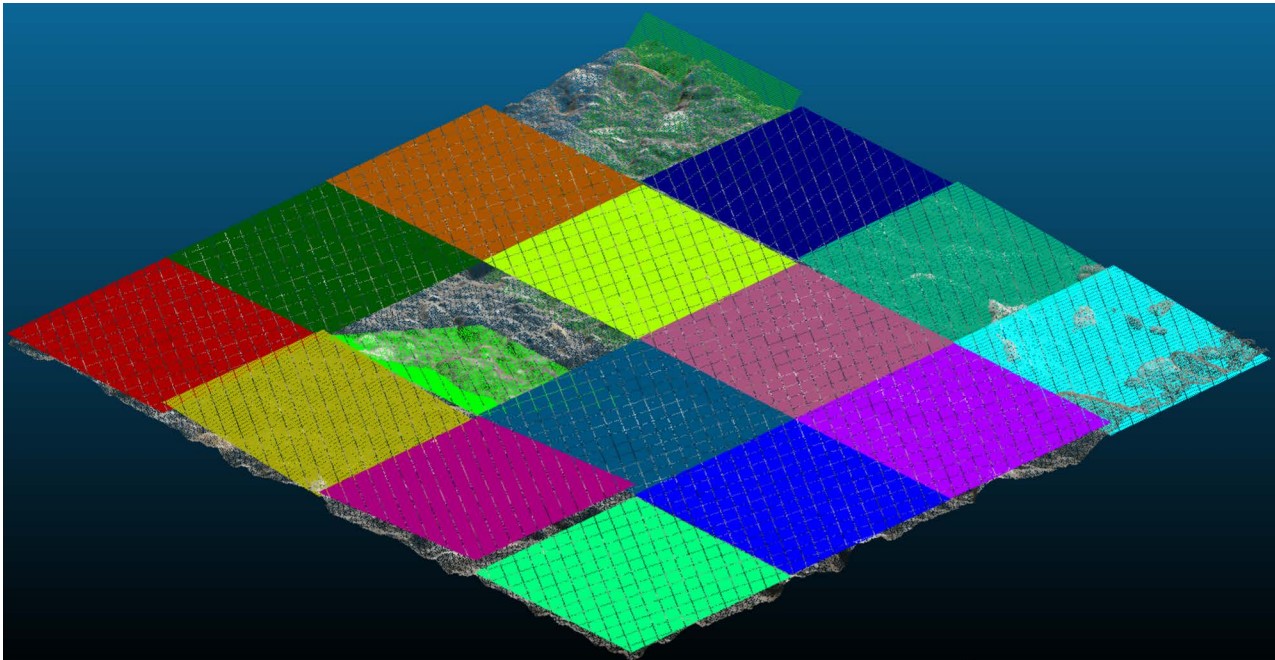

**Fig 27. A case study best-fit plane of 16 patches for three points.**

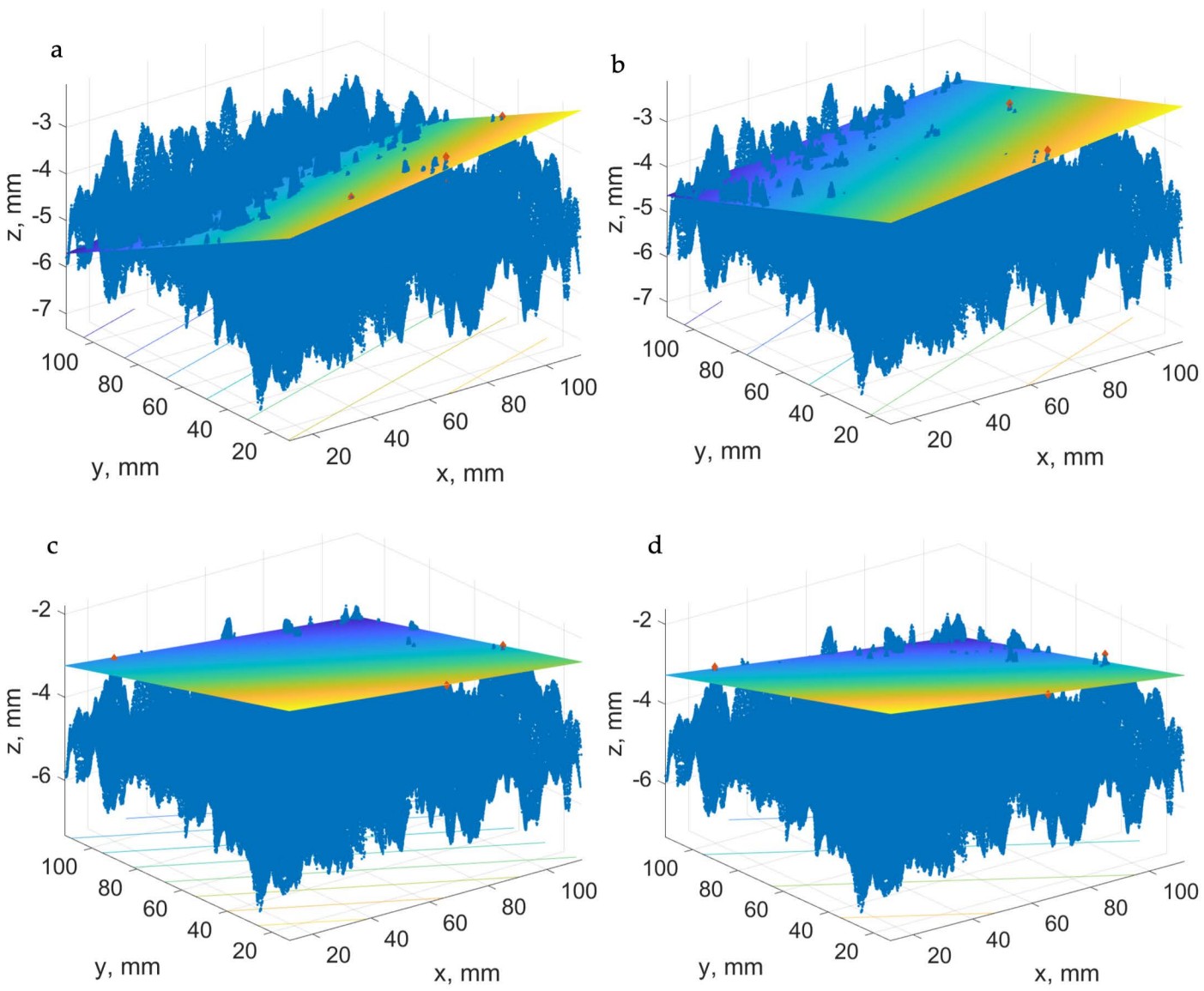

**Fig 28. A case study of the best-fit plane of four points with mask sizes** a) 5 mm, b) 10 mm, c) 15 mm, d) 20 mm.

The processing of the measured values was carried out in several ways. Their variations in the difference of the resulting values are often very small, but despite this, several basic findings can be established. Fig 32 shows the differences between the sand patch method and the different types of photogrammetric pavement investigation tests. It can be seen that the best displacements are obtained when the best fit plane method is used over a sufficiently small area. In this area it is also advisable to select multiple points with the highest height. According to Fig 32, A 07 and A 08 appear to be the best evaluation methods. Both of these cases have minimal errors compared to the sand patch method and can be considered very accurate. Similar quality results can also be seen in approaches A 15, A 09 and A 14. The reason for the quality of the results is obvious. There is a detailed area distribution and calculation with several values of the top heights in the image. In 3 of the 5 approaches mentioned, the best fitting method is used over the least squares method. In the remaining two cases, it involves

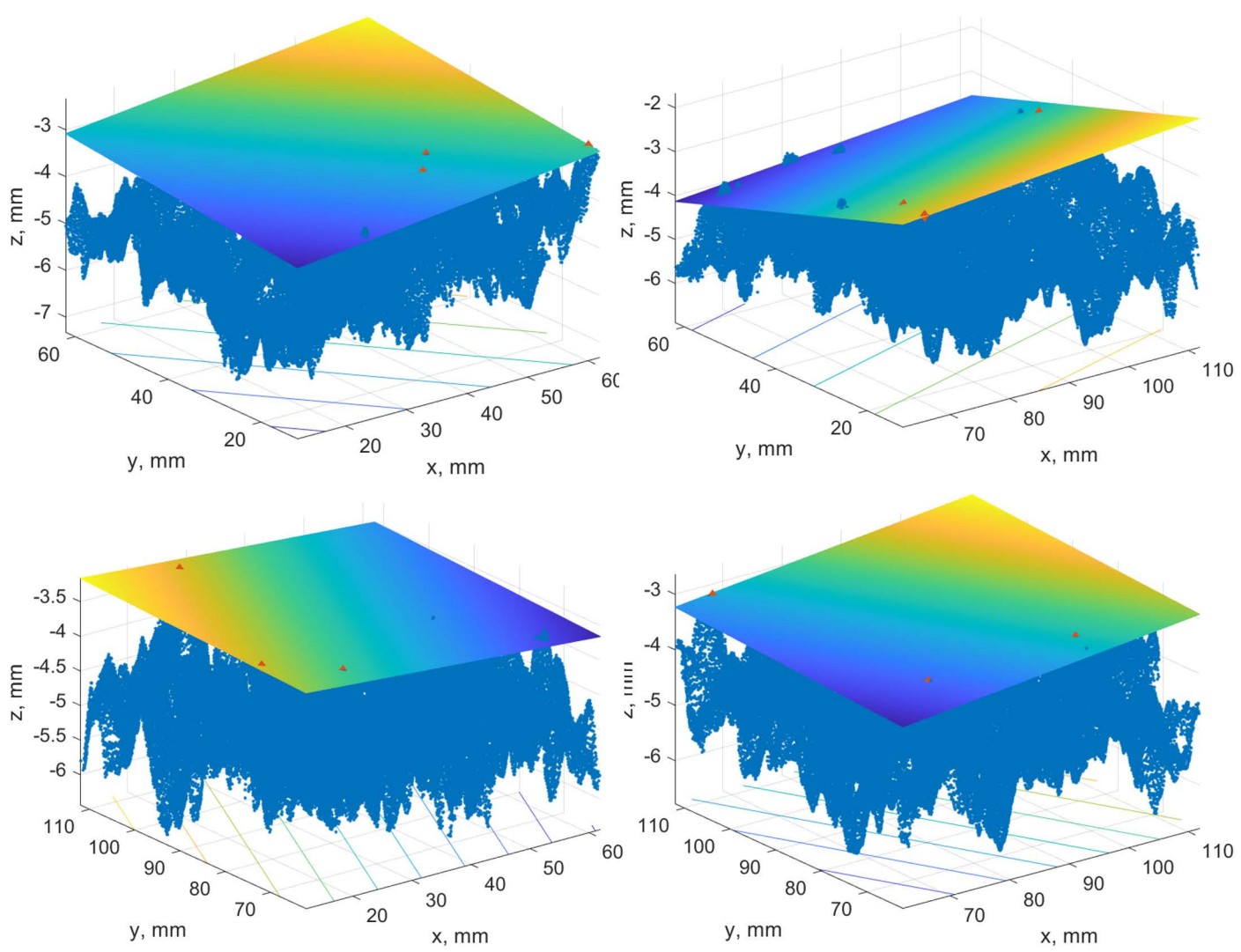

**Fig 29. A case study of the plane over 3 points for 4 patches with a mask sizes 2.5 mm.**

the use of horizontal plane determination over actual points and again using the approximation method (FoT).

The opposite example can be the gain of results realized through only one patch mask (A 16–A 19) using 3 points. The measured values directly show how crucial the plane determination method is. If a large area is covered, a larger variability in the heights can be observed. Only one significant roughness is needed and the whole plane can then be assessed according to this area.

However, if more points (A 03–A 06) are considered on the same patch size, as in the previous case, and are calculated over more points, it can be seen that the error rate is again reduced and the resulting accuracy is increased.

If measurements are made as part of other tests, it is advisable to perform them on a dry and clean surface. Dust and other debris may affect the macrotexture of the road surface and can lead to false detection of non-high points on the test surface. Consequently, the subsequent evaluation may not be performed correctly.

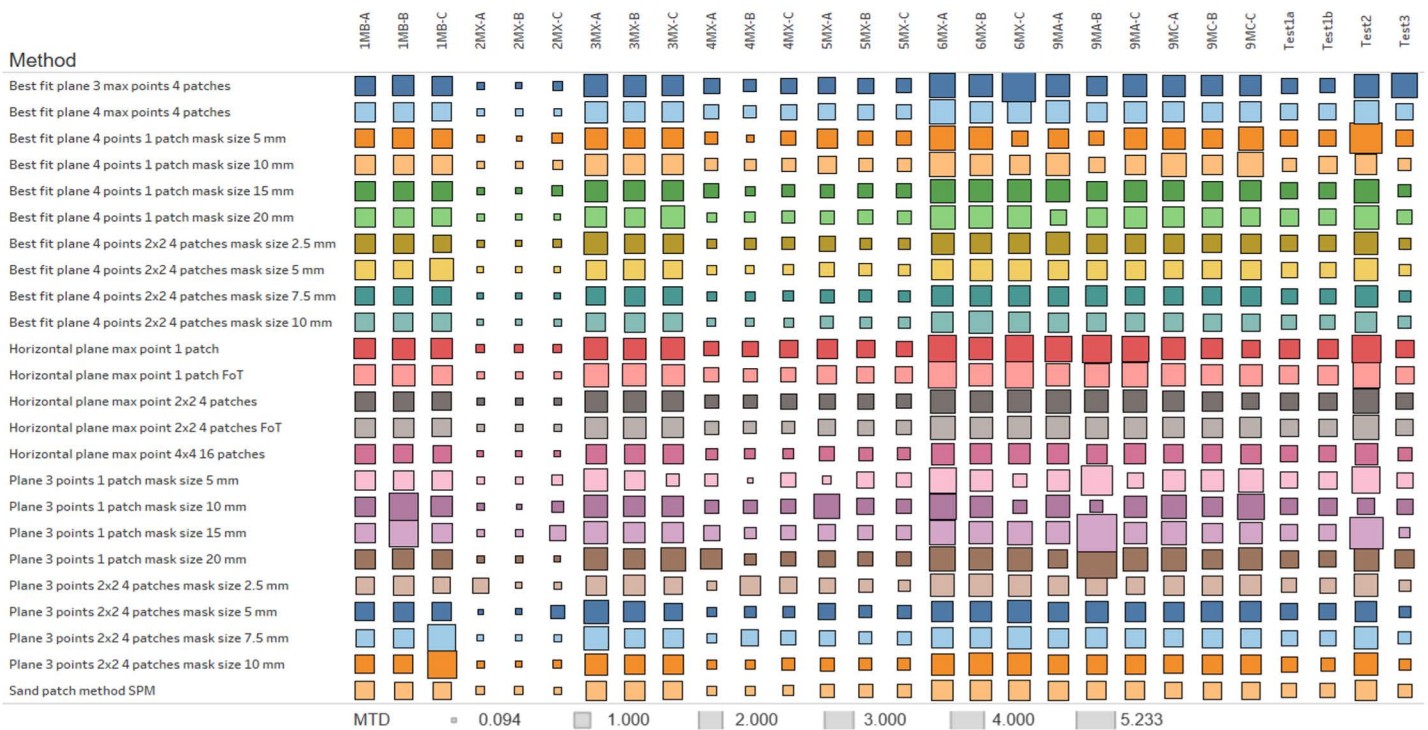

**Fig 30. Heatmap of MTD values for different calculation approaches.**

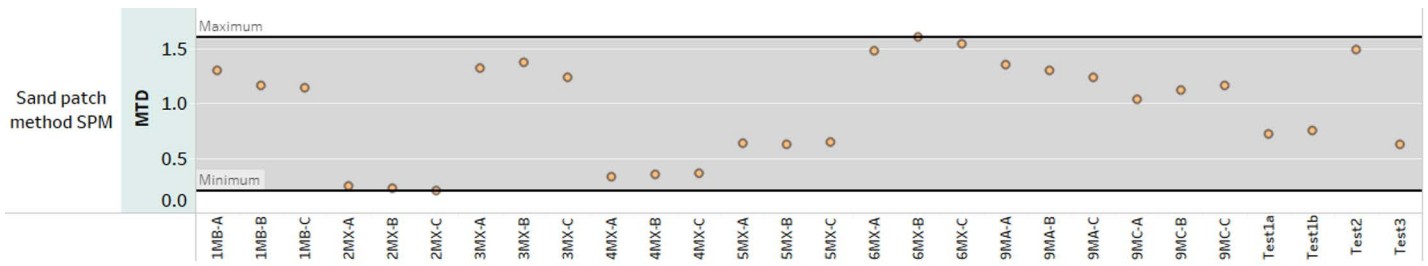

**Fig 31. MTD values for sand patch method.**

A water film on the road surface can cause measurement inaccuracies similar to those caused by dust particles, but it can also result in inappropriate light reflections that affect the acquisition of photographs necessary for properly creating a point cloud of the surface.

Similarly, it is recommended to assess the pavement sample in a location that is sufficiently illuminated. Alternatively, an external light source may be used. However, shading and contrast transitions should be avoided. Normalizing the lighting conditions improves the quality of the photographs, which, along with the aforementioned aspects, impacts the final result.

The results of the work were verified using the sand patch method defined in international standards. According to Fig 32, it can be seen that the Final Estimation produces very accurate results in several cases, with differences from the reference SPM being less than 0.2 in six instances.

| Approach | R2-score | Mean squared error [σ] | Mean absolute error [m] | Mean absolute percentage error [%] | Median absolute error [δ] | Max error [Δ] | Correlation vs. SPM [r] | Final estimation [E] |
|---|---|---|---|---|---|---|---|---|
| (A01) Best-fit plane 3 max points 4 patches | 0.63 | 0.22 | 0.30 | 0.31 | 0.22 | 1.74 | 0.79 | 0.42 |
| (A02) Best-fit plane 4 max points 4 patches | 0.90 | 0.03 | 0.15 | 0.22 | 0.14 | 0.42 | 0.95 | 0.21 |
| (A03) Best-fit plane 4 points 1 patch mask size 20 mm | 0.87 | 0.04 | 0.15 | 0.18 | 0.12 | 0.74 | 0.93 | 0.21 |
| (A04) Best-fit plane 4 points 1 patch mask size 15 mm | 0.92 | 0.02 | 0.13 | 0.19 | 0.13 | 0.36 | 0.96 | 0.19 |
| (A05) Best-fit plane 4 points 1 patch mask size 10 mm | 0.73 | 0.09 | 0.21 | 0.24 | 0.15 | 0.88 | 0.86 | 0.28 |
| (A06) Best-fit plane 4 points 1 patch mask size 5 mm | 0.59 | 0.20 | 0.29 | 0.38 | 0.17 | 1.56 | 0.77 | 0.40 |
| (A07) Best-fit plane 4 points 2x2 4 patches mask size 10 mm | 0.97 | 0.01 | 0.06 | 0.10 | 0.05 | 0.19 | 0.99 | 0.12 |
| (A08) Best-fit plane 4 points 2x2 4 patches mask size 7.5 mm | 0.96 | 0.01 | 0.08 | 0.12 | 0.06 | 0.23 | 0.98 | 0.14 |
| (A09) Best-fit plane 4 points 2x2 4 patches mask size 5 mm | 0.93 | 0.02 | 0.11 | 0.13 | 0.09 | 0.58 | 0.96 | 0.17 |
| (A10) Best-fit plane 4 points 2x2 4 patches mask size 2.5 mm | 0.90 | 0.03 | 0.14 | 0.17 | 0.15 | 0.39 | 0.95 | 0.20 |
| (A11) Horizontal plane max point 1 patch | 0.77 | 0.12 | 0.30 | 0.30 | 0.27 | 0.74 | 0.88 | 0.34 |
| (A12) Horizontal plane max point 1 patch FoT | 0.87 | 0.06 | 0.20 | 0.24 | 0.19 | 0.52 | 0.94 | 0.25 |
| (A13) Horizontal plane max point 2x2 4 patches | 0.89 | 0.04 | 0.16 | 0.23 | 0.16 | 0.51 | 0.94 | 0.22 |
| (A14) Horizontal plane max point 2x2 4 patches FoT | 0.94 | 0.02 | 0.12 | 0.19 | 0.11 | 0.30 | 0.97 | 0.18 |
| (A15) Horizontal plane max point 4x4 16 patches | 0.96 | 0.01 | 0.08 | 0.15 | 0.08 | 0.24 | 0.98 | 0.15 |
| (A16) Best-fit plane 3 points 1 patch mask size 20 mm | 0.42 | 0.51 | 0.40 | 0.37 | 0.26 | 3.31 | 0.65 | 0.66 |
| (A17) Best-fit plane 3 points 1 patch mask size 15 mm | 0.47 | 0.59 | 0.50 | 0.38 | 0.32 | 3.18 | 0.68 | 0.71 |
| (A18) Best-fit plane 3 points 1 patch mask size 10 mm | 0.35 | 0.31 | 0.38 | 0.54 | 0.16 | 1.32 | 0.59 | 0.52 |
| (A19) Best-fit plane 3 points 1 patch mask size 5 mm | 0.50 | 0.30 | 0.38 | 0.59 | 0.25 | 1.68 | 0.71 | 0.53 |
| (A20) Best-fit plane 3 points 2x2 4 patches mask size 10 mm | 0.79 | 0.08 | 0.13 | 0.14 | 0.07 | 1.39 | 0.89 | 0.24 |
| (A21) Best-fit plane 3 points 2x2 4 patches mask size 7.5 mm | 0.71 | 0.10 | 0.21 | 0.31 | 0.15 | 1.26 | 0.84 | 0.31 |
| (A22) Best-fit plane 3 points 2x2 4 patches mask size 5 mm | 0.87 | 0.04 | 0.16 | 0.26 | 0.16 | 0.48 | 0.93 | 0.23 |
| (A23) Best-fit plane 3 points 2x2 4 patches mask size 2.5 mm | 0.67 | 0.08 | 0.23 | 0.37 | 0.22 | 0.85 | 0.82 | 0.32 |

**Fig 32. Statistical measures for various approaches.**

It can be stated that the method which provides the most reliable results for the evaluation of the given photogrammetric measurements is the one based on the calculation of the plane via the best fit method using multiple height points on a sufficiently small mask area (7.5–10 mm). In conclusion, the required functionality and accuracy are verified and the pavement friction can be assessed in this way.

## Acknowledgments

No funding.

## Author contributions

**Conceptualization:** Zdeněk Svatý, Roman Shults, Luboš Nouzovský, Karel Kocián.

**Data curation:** Zdeněk Svatý, Roman Shults, Luboš Nouzovský, Tomáš Blodek.

**Formal analysis:** Zdeněk Svatý, Pavel Vrtal, Luboš Nouzovský.

**Funding acquisition:** Luboš Nouzovský.

**Investigation:** Roman Shults, Tomáš Kohout.

**Methodology:** Roman Shults.

**Project administration:** Zdeněk Svatý, Pavel Vrtal, Luboš Nouzovský.

**Resources:** Pavel Vrtal, Tomáš Kohout, Tomáš Blodek.

**Software:** Roman Shults.

**Supervision:** Zdeněk Svatý, Karel Kocián.

**Validation:** Zdeněk Svatý, Roman Shults, Tomáš Kohout, Karel Kocián.

**Visualization:** Tomáš Kohout, Tomáš Blodek.

**Writing – original draft:** Pavel Vrtal, Luboš Nouzovský, Karel Kocián.

**Writing – review & editing:** Pavel Vrtal.

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
