## [Decision Letter · Decision Letter 0]

8 Dec 2024

PONE-D-24-46481Photogrammetric Approach to Detect Road Pavement FrictionPLOS ONE

Dear Dr. Vrtal,

Thank you for submitting your manuscript to PLOS ONE. After careful consideration, we feel that it has merit but does not fully meet PLOS ONE’s publication criteria as it currently stands. Therefore, we invite you to submit a revised version of the manuscript that addresses the points raised during the review process.

We look forward to receiving your revised manuscript.

Kind regards,

Jiaolong Ren

Academic Editor

PLOS ONE

Journal Requirements:

4. Please ensure that you refer to Figure 8 and Figure 9 in your text as, if accepted, production will need this reference to link the reader to the figure.

5. Please include a caption for figure 34.

Reviewers' comments:

Reviewer's Responses to Questions

**Comments to the Author**

1. Is the manuscript technically sound, and do the data support the conclusions?

Reviewer #1: Yes

Reviewer #2: Yes

2. Has the statistical analysis been performed appropriately and rigorously? 

Reviewer #1: Yes

Reviewer #2: Yes

3. Have the authors made all data underlying the findings in their manuscript fully available?

Reviewer #1: Yes

Reviewer #2: Yes

4. Is the manuscript presented in an intelligible fashion and written in standard English?

Reviewer #1: Yes

Reviewer #2: Yes

5. Review Comments to the Author

Reviewer #1: After a comprehensive review of the research titled “Photogrammetric Approach to Detect Road Pavement Friction” submitted by the researchers and sent by your respected journal, the researcher reached the following:

1. In terms of the scientific idea, the research is good in the field of methods and presents a new idea in the working mechanism.

2. From a philosophical and literary perspective, the research contains very little of the proposition. Despite its importance in the research, it can be summarized as in a paragraph Current Knowledge line 72.

3. Are the twenty-three different approaches to evaluation based on standards or just studies? Please confirm line 43.

4. Recommendation to accept the research after making minor and pre-determined modifications.

Thanks for the great research and new ideas

Good luck

Reviewer #2: 1. problem statement not clear in the abstract

2. The conclusions are poor, need to rewrite

3. Add recommendations

4. Add ref to some equations mentioned in paper

5. Add limitation for this work

6. the man gaps should mention and clarify according to previous related works.

6. PLOS authors have the option to publish the peer review history of their article (what does this mean? ). If published, this will include your full peer review and any attached files.

**Do you want your identity to be public for this peer review?** For information about this choice, including consent withdrawal, please see our Privacy Policy .

Reviewer #1: No

Reviewer #2: No

---

## [Author Response · Author response to Decision Letter 1]

3 Jan 2025

Reviewer #1:

1. In terms of the scientific idea, the research is good in the field of methods and presents a new idea in the working mechanism.

• Thank you!

2. From a philosophical and literary perspective, the research contains very little of the proposition. Despite its importance in the research, it can be summarized as in a paragraph Current Knowledge line 72.

• The main source for this research is Luhmann et al (Close-Range Photogrammetry and 3D Imaging) and the standards [5] ČSN EN 13036-4:2012 (73 6177) Road and airfield surface characteristics - Test methods - Part 4: Method for measurement of slip/skid resistance of a surface: The pendulum test. Most of the necessary information can be found in this literature. As the work is focused on an innovative method using a specific procedure, the availability of publications by other authors is very limited. From the point of view of the author's team, the use of existing references appears to be sufficient.

3. Are the twenty-three different approaches to evaluation based on standards or just studies? Please confirm line 43.

• The abstract has been modified to clarify that the 23 different surface evaluation approaches were determined based on the authors' experience during the measurements. In this context, standards were utilized to define the road surface samples.

4. Recommendation to accept the research after making minor and pre-determined modifications.

Thanks for the great research and new ideas.

• Thank you!

Reviewer #2:

1. problem statement not clear in the abstract

• The abstract has been modified to explain the aim of the thesis and its progress more clearly.

2. The conclusions are poor, need to rewrite

• Based on the submission guidelines of the journal, it was decided to combine the Discussion and Conclusion into one chapter. Only the Discussion is now included, where the authors believe that all the important conclusions of the paper are mentioned. At the same time, recommendations how to repeat the measurements have been added. At the same time, limiting factors that might limit the thesis have also been added.

3. Add recommendations

• If the authors have understood the question correctly, there is a section in the Discussion recommending how to make similar measurements.

4. Add ref to some equations mentioned in paper

• Has been added to the Manuscript.

5. Add limitation for this work

• Has been added to the Discussion.

6. the main gaps should mention and clarify according to previous related works.

• As the method of measurement is innovative, the main limitations cannot be sufficiently compared with previous work related to other studies. Comparisons have been made according to official standards dealing with pavement friction. These standards define how measurements can be made using the sand method and what results can be achieved (EN 13036-4:2012 (73 6177) Road and airfield surface characteristics - Test methods - Part 4: Method for measurement of slip/skid resistance of a surface: The pendulum test). Based on Luhmann et al (Close-Range Photogrammetry and 3D Imaging) it was possible to determine the necessary conditions for proper processing of the photogrammetric measurement. Then, the results show that the accuracy is sufficiently high and the measurement method can be considered as verified and valid.

---

## [Decision Letter · Decision Letter 1]

13 Jan 2025

Photogrammetric Approach to Detect Road Pavement Friction

PONE-D-24-46481R1

Dear Dr. Pavel Vrtal,

We’re pleased to inform you that your manuscript has been judged scientifically suitable for publication and will be formally accepted for publication once it meets all outstanding technical requirements.

Kind regards,

Jiaolong Ren

Academic Editor

PLOS ONE

Additional Editor Comments (optional):

Reviewers' comments:

Reviewer's Responses to Questions

**Comments to the Author**

1. If the authors have adequately addressed your comments raised in a previous round of review and you feel that this manuscript is now acceptable for publication, you may indicate that here to bypass the “Comments to the Author” section, enter your conflict of interest statement in the “Confidential to Editor” section, and submit your "Accept" recommendation.

Reviewer #1: All comments have been addressed

Reviewer #2: All comments have been addressed

2. Is the manuscript technically sound, and do the data support the conclusions?

Reviewer #1: Yes

Reviewer #2: Yes

3. Has the statistical analysis been performed appropriately and rigorously? 

Reviewer #1: Yes

Reviewer #2: N/A

4. Have the authors made all data underlying the findings in their manuscript fully available?

Reviewer #1: Yes

Reviewer #2: (No Response)

5. Is the manuscript presented in an intelligible fashion and written in standard English?

Reviewer #1: Yes

Reviewer #2: Yes

6. Review Comments to the Author

Reviewer #1: After reviewing the research paper with the title Photogrammetric Approach to Detect Road Pavement Friction and Number PONE-D-24-46481, it became clear to the reviewer that the research paper is very valuable in terms of the idea and the way it was presented and written. Therefore, we recommend accepting the research after making minor modifications to it, as shown:

1. The introduction is brief. Please rephrase it, preferably adding a simple definition of the study, results and conclusions.

2. The sources are considered old in terms of history and not the scientific content. It is better to add modern sources.

3. Please increase the resolution of the images installed in the search.

Thank you for the good work. I wish you success.

Reviewer #2: Good work, but need to add main conclusion's recommendation for future work

And what are the conclusions you may recommend from this work to be helpful for field work

7. PLOS authors have the option to publish the peer review history of their article (what does this mean? ). If published, this will include your full peer review and any attached files.

**Do you want your identity to be public for this peer review?** For information about this choice, including consent withdrawal, please see our Privacy Policy .

Reviewer #1: **Yes: ** Rana Amir Yousif

Reviewer #2: No

---

## [Editor Report · Acceptance letter]

PONE-D-24-46481R1

PLOS ONE

Dear Dr. Vrtal,

I'm pleased to inform you that your manuscript has been deemed suitable for publication in PLOS ONE. Congratulations! Your manuscript is now being handed over to our production team.

Kind regards,

on behalf of

Dr. Jiaolong Ren

Academic Editor

PLOS ONE